# Acoustic monitoring reveals a diel rhythm of an arctic seabird colony (little auk, *Alle alle*)
Evgeny A. Podolskiy [1] ✉, Monica Ogawa[2], Jean-Baptiste Thiebot[3], Kasper L. Johansen [4] & Anders Mosbech[4]

The child-like question of why birds sing in the morning is difficult to answer, especially in polar regions. There, in summer animals live without the time constraints of daylight, and little is known about the rhythmicity of their routines. Moreover, in situ monitoring of animal behavior in remote areas is challenging and rare. Here, we use audio data from Greenland to show that a colony of a key Arctic-breeding seabird, the little auk (*Alle alle*), erupts with acoustic excitement at night in August, under the midnight sun. We demonstrate that the acoustic activity cycle is consistent with previous direct observations of the feeding and attendance patterns of the little auk. We interpret this pattern as reflecting their foraging activities, but further investigation on fledging and predators is needed. The study demonstrates that acoustic monitoring is a promising alternative to otherwise demanding manual observations of bird colonies in remote Arctic areas.

The only study on the sounds of the little auk (*Alle alle*) in Greenland[1] is scientifically inspiring and emotionally moving for two reasons. First, Ferdinand[1] observed that at a distance the voices of birds were heard as something between the protracted scream of a gull and the hoot of a deep siren and that it was an experience and a sight of such beauty and intensity as to defy description; and, second, that nocturnal activity of the little auks was known to the Inuit. Whether polar seabirds have diel acoustic behaviors in continuous daylight remains unknown, but could be investigated using passive acoustic monitoring. Records of animal sounds are becoming increasingly important behavioral and ecological indicators[2,3]. Acoustic records of animal calls are collected as a substitute for demanding direct observations, and robust recorders are widely available[4]. Advances in hardware and software (e.g., artificial intelligence) have opened new avenues for assessing biodiversity, identifying behavioral patterns, detecting changes, and comparing them across time, space, and taxa[5–8]. Although passive acoustic monitoring of bird colonies and acoustic-based bird-density estimation are increasingly being employed[9–14], it has not been reported in polar regions but is necessary as these are undergoing the most unprecedented environmental transition in human history. Furthermore, how birds time their behavior is a fundamental scientific question because it is relevant to all aspects of avian biology, including social behavior, reproduction, foraging, migration, orientation, and vocalization[15]. However, it has rarely been

investigated in polar environments by acoustic methods. In mid-latitudes, a dawn chorus might be obvious to casual observers and appear to be related to a combination of factors[15,16]. In high latitudes, the midnight sun attenuates temporal constraints on the animals' activity, may lead to free-running-like patterns[17], and thus offers a unique opportunity to study circadian biology[18] and examine the existence of a diel acoustic rhythm under continuous daylight.

The little auk (also known as the dovekie) is a small seabird endemic to the Arctic with an important engineering role in structuring Arctic marine and terrestrial ecosystems[19]. This small diving species is the most abundant seabird in the North Atlantic, which provides marine-derived nutrients (and acidity) to terrestrial ecosystems, enhances primary production, and truncates freshwater food webs[19]. Today, little auk is a subject of intense study as a sentinel of accelerating ecological changes in the Arctic[20]. The most important breeding area of the little auk (~30 million pairs in total) is located in Northwest Greenland[20–22]. There, they have been an important part of local subsistence hunting for millennia (e.g., for the preparation of the traditional kiviaq), have influenced Inuit settlement patterns[23], and are culturally relevant to the Inuit[24–26].

Although the little auk is one of the most abundant birds in the Arctic, studies of its vocalization are extremely limited. To our knowledge, in Greenland, only 1.5 h recordings of their calls made before chick-rearing

[1]Arctic Research Center, Hokkaido University, Sapporo, Japan. [2]Graduate School of Environmental Science, Hokkaido University, Sapporo, Japan. [3]Graduate School of Fisheries Sciences, Hokkaido University, Hakodate, Japan. [4]Department of Ecoscience, Aarhus University, Roskilde, Denmark. ✉e-mail: evgeniy.podolskiy@gmail.com

have been reported[1]. The only digital-era recordings, several hours each, have been collected in Spitsbergen[27,28], together with several short audios from Norway deposited online (https://xeno-canto.org/species/Alle-alle). It has also been suggested that the diving noise of the little auk (1–4 kHz) can be detected on hydrophone records[29]. There are also some older verbal descriptions in different languages. According to Ferdinand[1], the voice of the little auk is described very differently by the various authors (e.g., pirrr rirrr rirrr – trrr trrr tet tet tet trrrr – gii gii gii – kriiiik iiiik ak ak ak ak ak ak).

The above examples are only qualitative and highlight the desperate need for more quantitative and modern approaches. While a lack of terrestrial soundscape research in the tropics has been recognized as a crucial gap[14], Arctic terrestrial soundscapes remain nonexistent as a topic[30]. In this study, we aimed to fill this current knowledge gap by collecting continuous records of the sound generated by a little auk colony in Greenland. Using this approach, we test for the existence of a circadian-like rhythm in the colony's sound activity. Our study (1) shows a remarkable diel pattern as the colony explodes with excitement every night, despite the continuous daylight, (2) demonstrates that the ambient sound may help to indirectly infer activity of a polar colony, and (3) generates the non-interrupted dataset for this species, which is underrepresented in the literature (and can be re-analyzed for understanding vocalization repertoire and as methods progress to allow reliable detection of overlapping calls).

## Results and discussion
### Soundscape variability
Continuous audio data were obtained during chick-rearing period at two recording sites, near Siorapaluk (July; post-hatch stage) and in the colony (August; around fledging stage) (Fig. 1). In order to identify bird-related variation in the soundscape, we used four different complementary approaches (as detailed below and in Methods). The data were: first, visualized as long-term spectrograms (LTSs) and represented as relative sound intensity, RSI (Figs. 2, 3); second, aurally inspected; third, presented as commonly used biophonic indices (Supplementary Fig. 1); and finally, fourth, decomposed into main types of sounds via supervised semi-automatic signal detection and clustering (Supplementary Fig. 2).

Figure 2 demonstrates the temporal variation in the frequency and amplitude of the soundscape, together with the corresponding power spectral densities (Fig. 2b, d). Both study sites had similar frequency peaks at 2, 6.7, and ~16.6 kHz. Considering the manufacturer's description of the sound recorder, we interpreted the 6.7 and 16.6 kHz peaks as the instrument's higher sensitivity and self-noise, respectively, and the peak at 2 kHz as ambient noise (of biological origin, as shown below).

To understand what sounds dominated the soundscape at the colony, we manually inspected and listened to several (~15) arbitrarily chosen audio files (Fig. 2e–g). In addition to previously recognized individual thrilling and single calls[1], we found that (i) at around 2 kHz, a continuous chorus of the colony could be heard (defined as many individuals making sounds that overlap or are emitted in rapid succession[31]); (ii) wing flapping was most distinct at around 7 kHz; and (iii) the power of the little auk's vocalizations was concentrated mainly below 10 kHz, with some harmonics leaking into higher frequencies.

The July data collected near the village showed no obvious temporal variation and some rare high-amplitude transient signals (Fig. 2a). On average, the hourly number of detected sounds, was about six times lower than in the colony (see Methods). Nevertheless, visual and aural data inspection confirmed the continuous but weak presence of bird sounds (suggesting that the vocalizations were of insufficient amplitude to dominate the ambient noise because of the distance involved). On the contrary, the August data recorded at the colony had a strong diurnal rhythm (Fig. 2c). Spectra and individual calls were broadband (Fig. 2c–g) in an area with a negligible presence of non-target species around the recorder, which otherwise could be included in the soundscape[14]. Therefore, for convenience, we presented the data as a broadband median relative sound intensity time series (Fig. 3). In general, a similar logarithmic measure of the effective pressure of a sound (sound pressure level) is a commonly used indicator of bioacoustic activity[32,33].

In Fig. 3, we observe the following three main features. First, there is a clear diurnal pattern, with maximum intensity of sound after midnight (02:30–03:30) and minimum in the afternoon (14:00–16:00). Second, the noise lags behind the elevation of the sun by 120 ± 45 min (mean and

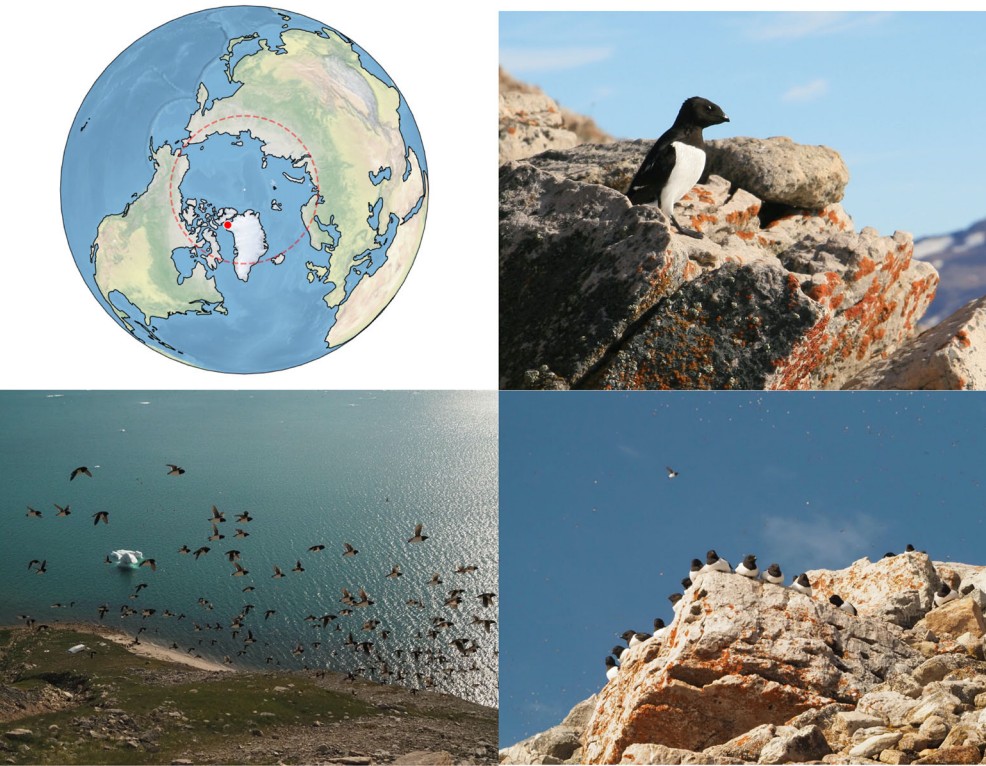

**Fig. 1 | Study site.** Little auk colony near Siorapaluk, Northwest Greenland (Photo: M. Ogawa, August 2022; E. Podolskiy, July 2016). The Arctic circle is shown in red.

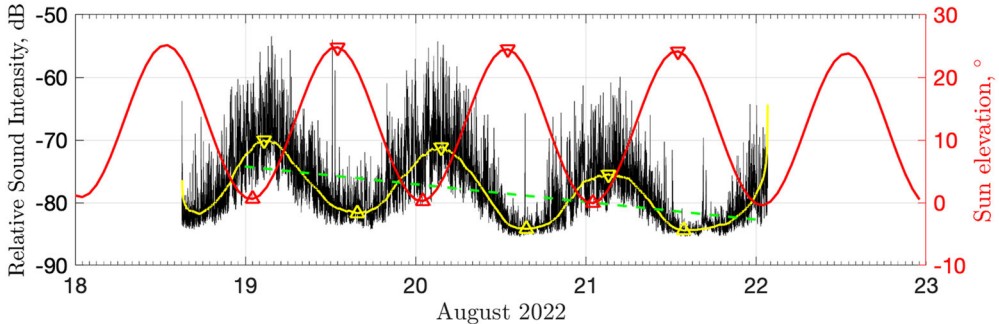

**Fig. 2 | Ambient sound spectra and little auk sounds. a, c** Long-term spectrograms of sound recorded near the village in July and at the colony in August 2022 (Siorapaluk, Greenland) with (**b, d**) the corresponding probability distribution of power spectral densities (computed with 10 s-long sliding windows; red curves show percentiles levels). Typical bird's sounds at the little auk colony near Siorapaluk, Greenland: (**e**) Trilling call (with motiv I--III); (**f**) single call; and (**g**) wing flapping. Note the apparent repetition of wing flapping every 18 s, probably indicative of the birds circling repeatedly above the colony. Source data underlying the plots (**a–g**) are available in refs. 55,56.

**Fig. 3 | Noise and sun.** Median relative sound intensity (RSI; black) at the little auk colony near Siorapaluk, Greenland (local time, 2022), compared with sun elevation (red). Yellow color corresponds to the filtered time series (we applied a one-dimensional median filter to smooth the signal using a 5 h-long sliding window). Triangles mark automatically detected peaks and troughs to highlight a lag. Green dashed line shows the trend of the yellow curve (excluding marginal data on 18 and 22 August). Source data underlying the graph are available in refs. 55,56 (for sound), and in https://www.sunearthtools.com/dp/tools/pos_sun.php (for the sun elevation).

standard deviation values for six peak–trough pairs marked in Fig. 3). Third, the noise decreases in amplitude with time (2.8 dB per day; Mann-Kendall test rejects the null hypothesis of trend absence at the alpha significance level of 0.01; *p*-value < 0.001).

Audio files from ~03:00 and ~15:00 (separated by 12 h) demonstrated a clearly perceivable difference in the level of sound excitement in the colony (Supplementary Audio 1 and 2). This tendency was quantitatively re-confirmed using independent biometrics of acoustic activity: both

commonly employed biophonic indices (see Methods) were low at daytime (Supplementary Fig. 1). Hourly detection rates of the main sound classes showed a similar tendency (Supplementary Fig. 2). In particular, all sounds produced by little auks—including vocalizations (single calls and trilling calls) and wing flapping—showed minima at daytime and were detected more frequently between evening and morning. The above-mentioned results independently highlight that little-auk sounds dominated the soundscape at the colony, which, therefore, may contain some behavioral information.

We plotted our filtered noise-intensity data versus previous reports of attendance, chick feeding, and zooplankton availability (Fig. 4). These data show that the nocturnal increase in noise is consistent with the chick feeding and attendance patterns of the little auk.

### Diel acoustic pattern and its potential drivers

In general, daily activity rhythms are surprisingly diverse in Arctic-breeding birds[18]. Our results are consistent with the diurnal rhythm of neighboring little auk colonies in Qoororsuaq (Fig. 4b) and on Cape Atholl (also near Thule), visually noticed by Ferdinand[1]. Ferdinand wrote that in June, the birds came ashore one or two hours after midnight, and then there was great activity during the whole night and early in the morning (a fact which was known to the Greenlanders). This behavior reminds another species of small non-arctic seabirds: Leach's storm-petrels (*Hydrobates leucorhous*) are nocturnally active and highly vocal in their breeding colonies[10,12]. In particular, increased abundance of Leach's storm-petrels was highlighted by

significantly related acoustic and radar data, which also found a decrease in acoustic activity on full-moon nights, presumably due to predator avoidance[12].

Quantitative attendance studies of little auk colonies have been undertaken in Western Greenland (Horse Head)[34], Northwestern Greenland (Qoororsuaq[35]; and this study), and Western Svalbard (Hornsund and Magdalenefjorden)[36,37] and have shown similar rhythmicity. However, our comparative review of these studies (provided in Supplementary Note 1 and summarized in Table 1) highlighted that the literature has no consensus on the cause of this diurnal rhythm in little auk behavior, which, moreover, can drift depending on the phenological stage (e.g., Supplementary Fig. 3). Table 1 shows that there are at least three alternative hypothesizes, one related to feeding, another related to predation pressure modulating the attendance, and the third related to fledging, as further discussed below (and in Supplementary Note 1). Our review also reveals that there is no discussion of the link between the number of feeds to nestling chicks and the number of birds on land (i.e., if there are more birds, one would expect more feeds, as reported in[34], but not found in[37]).

Considering the biological rhythmicity of Fig. 4, it makes sense to expect an overall increase in acoustic activity of the colony due to a high colony attendance after midnight. However, little auks may attend the colony for different reasons depending on breeding phenology.

For instance, the most detailed in situ observations on chick meals in Greenland suggest intense feeding by many returning birds at night, and calm conditions in the daytime as the birds leave for the sea (Fig. 4a). Here,

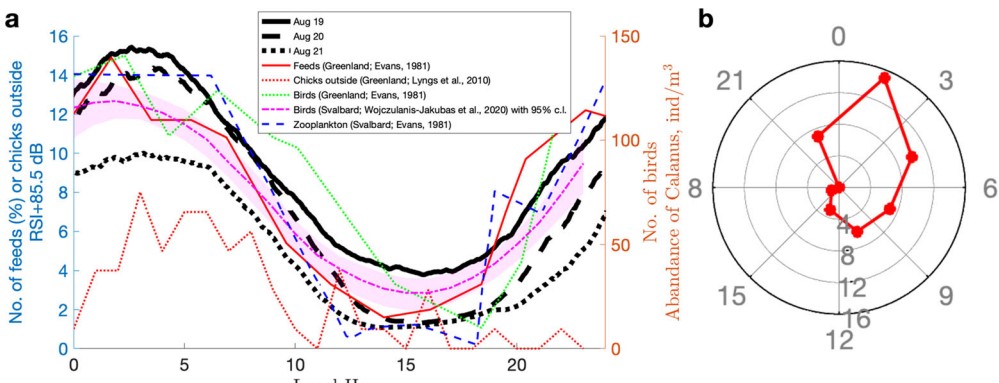

**Fig. 4 | Rhythmicity in this and other studies. a** Diel variation in colony noise (black; 19–21 August 2022; RSI+85.5 dB for better visibility) compared with observations from the literature for Greenland and Western Svalbard[34,35,37]: (1) average number of little auks on land in Greenland (green; 31 July–10 August 1974[34]) and in Svalbard (magenta; end of July–beginning of August 2009 and 2010; scaled by 4 for better visibility[37]; shading shows 95% confidence limits); (2) percentage of the total number of feeds to nestling little auks in 24 h (red; 29 July–10 August 1974[34]); (3) numbers of *Calanus* counted in 1 m³ sampled near the surface (blue; July–August[34]) (4) number of chicks recorded by camera (per hour) outside their nests (red dots; 31 July–15 August 2010[35]). **b** Peak occurrence hour of birds sitting on the rocks in the plot area as recorded by camera in Qoororsuaq, Northwestern Greenland (*n* = 49 days; bins are 3-h long; 31 July–23 August 2013, 2014, 2016). Raw data of bird counts in Qoororsuaq are shown in Supplementary Fig. 3 and available as Supplementary Data 1. Source data for black curves are taken from Fig. 3 and available in refs. [55,56]; values from other regions are available in refs. 34,35,37 and available as Supplementary Data 2.

**Table 1 | Possible drivers of a diel attendance cycle of the little auks according to previous studies ("o" – likely; "x" – unlikely; "–" – not discussed; details are provided in Supplementary Note 1)**

| Study | Location | Feeding | Predators | Fledging | Method to reveal diurnal rhythms |
|---|---|---|---|---|---|
| Evans[34] | W. Greenland | o | – | – | little auks visual count and chick weighting |
| Stempniewicz[36] | W. Svalbard | o | x | – | little auks and glaucous gull visual count |
| Wojczulanis-Jakubas et al.[37] | W. Svalbard | x | o | – | little auks visual count |
| Lyngs et al.[35] | NW Greenland | – | – | o | little auks visual/camera count |

The only surveys on the diel activity of little auks in Greenland[34,35] suggested that, first, the feeding cycle matches the availability of zooplankton, and, second, chicks are active outside nests at night. In Svalbard, Stempniewicz[36] counted gulls but concluded that their presence was driven by the little auk attendance, not vice versa; Wojczulanis-Jakubas[37] detected no feeding cycle and suggested that the zooplankton cycle was insignificant, and without counting gulls, hypothesized that it is safer at "night" for little auks.

one could suggest that arrival and departure of parents might lead to two sound peaks (i.e., "hi" and "bye"). However, this hypothesis would be relevant only for a simultaneous movement of all birds, who stay silent in the colony, and an assumption that arrival and departure have equivalent loudness. This is unlikely[1,24,34], especially when repeated flights during the night characterize such small and agile fliers as Leach's storm-petrels[12] and can be expected for little auks (wing flapping of which was reduced only in the afternoon; Supplementary Fig. 2)[35].

An alternative interpretation might be a nocturnal fledging of chicks. It was noted that at the end of the breeding season, the departure of little auk young—the main objects of predation—usually takes place at night hours[35,36]. For example, it was observed that during nearly two weeks of fledging period (3–18 August 2010) in Qoororsuaq, Northwest Greenland[35], fledging activity started around 2 am and stopped at 5 am. Similarly, the highest number of chicks outside nests (Fig. 4a) was also recorded between 1 am and 8 am[35]. However, to our knowledge, it remains unclear whether fledging timing is driven by nocturnal foraging opportunities or predation pressure, and if the chicks are outside their nests because adult birds are back.

A comparison of our results with the literature is not always straightforward. Given the differences between the studies in (1) the geographic locations of colonies, (2) the foraging distances (from 2.5 km up to ca. 100 kilometers), and (3) the survey dates (e.g., the timing of the breeding stages may vary slightly from year to year), we will limit our discussion of the fine time delays present in the data (Fig. 4) to avoid their over-interpretation.

It is debatable whether the diel attendance rhythm is entrained to the light cycle via foraging during the hours when the prey is closer to the sea surface or via predation pressure (Table 1, Supplementary Note 1).

On the one hand, the phenomenon of diel vertical migration (DVM) of zooplankton during the midnight sun has been repeatedly documented using echosounders across the Arctic Ocean[38,39], but remains underestimated in polar ornithological literature[37,40]. On the other hand, nocturnal activity of non-arctic Leach's storm-petrels has been associated with predator avoidance[12]. The systematic lag of ~2.5 h in the noise relative to the elevation of the sun was detected in this and a previous study[37]. In the present study, it is reasonable to suggest that the birds needed time to return to the colony from remote foraging grounds, which may have given rise to the observed lag. We cannot exclude the possibility that interference by predators was involved (see Supplementary Note 2, Supplementary Fig. 4). However, the resemblance of the feeding, attendance, and noise cycles, together with the lack of any record of a lag between attendance and predators in the literature[36,37], means that zooplankton remains our main candidate driver. Nocturnal fledging cannot be excluded from explanatory factors, but has yet to be confirmed quantitatively.

The strength of the biological clock has also been interpreted in terms of foraging in other Arctic species of birds, herbivores, and whales, but not in all birds, because temperature and mating strategies may be alternative drivers[18,41]. Furthermore, the rhythm of breeding-site attendance can be sex-specific in such Arctic seabirds as thick-billed murres[40]. In continuous light conditions of Antarctica, a diurnal rhythm of departure of adult Adelie Penguins was characterized by a higher numbers of birds leaving after midnight during the late chick-hatching period[42]. For chick-rearing emperor penguins, such period was not detected[43]. If the observed pattern of the little auk activity is related to foraging, then it is similar to the foraging pattern of the Greenlandic whales. Narwhal (*Monodon monoceros*) is more active when the sun is low[41]; and bowhead whale (*Balaena mysticetus*), is diving deeper in daytime to feed on the same copepods as the little auk[44].

If noise (Fig. 4a) is proportional to the overall number of birds (not only those rearing chicks), this might imply that many birds are out foraging exactly when food availability is minimal. However, energy demands of traveling to the foraging ground, shorter trips at night, and the requirement for sleep (which mysteriously does not occur on land, according to existing accounts) could explain these discrepancies.

The gradual reduction in the noise amplitude over 3 days is statistically significant, yet too short to draw any conclusions. However, it makes sense when we consider the gradual departure of the birds away from the breeding colonies at the end of the season, as shown for Qoororsuaq in Supplementary Fig. 3 (also see Fig. 1e in refs. [36] and [24]).

## Overall sound features

The detailed association between the ethological context and the sounds produced by adult birds has been examined at the beginning of the breeding season in Greenland (before incubation)[1] and during mating and incubation periods in Svalbard[27], and may differ during periods with chicks. Ferdinand[1] suggested that little auks produce some of the loudest sounds among members of the family Alcidae and attempted to systematize his observations to distinguish the following vocalizations – calls: single, trilling, flock singing; and pair calls: aggressive, clucking, and snarling. For example, a snarling sound was related to a pair searching for a nest, and the most common trilling call was used by flying and sitting birds. Based on manual inspection, Osiecka et al.[27] appended these natural vocalization types with flock's terror, near-nest's short call, and short trill (the former was attributed to unspecified predator's presence on the ground, and the latter two were often a part of a bout with Ferdinand's vocalization types). This led to a proposal that little auks might have a larger repertoire than other described alcids[27]. Ferdinand[1] suggested that the trilling call and single short call (Fig. 2e) were most common, and that most vocalizations were composed of variations of two main units. As shown below, our automatic analysis (see Methods) confirms his subjective judgment.

Among clean, non-overlapping detections, single calls were the most abundant ($n = 20,547$) short broad-band sounds with duration $0.09 \pm 0.01$ s, mean frequency of $3.7 \pm 0.3$ kHz, and up to 5 harmonics under 10 kHz. Trilling calls ($n = 110$) have a higher duration ($0.6 \pm 0.14$ s) and a mean frequency of $2.0 \pm 0.2$ kHz. Such calls are also broad-band with up to 10 harmonics under 10 kHz. A single wing flap sound ($n=139$) lasts for $0.85 \pm 0.27$ s and has a prominent mean frequency of $6.7 \pm 0.08$ kHz (a peak of nearly equivalent amplitude also exists at 1.5 kHz but generally harder to use for detection due to overlaps with other signals). In general, all these typical sounds were detected rarely in daytime, in line with our continuous soundscape analysis (Supplementary Fig. 2). However, considering the limitations of signal-detection methods in saturated soundscapes[12], it remains difficult to interpret fine differences in the number of detected sounds during the most noisy period (from evening till morning hours). Nevertheless, taking into account that trilling calls are the most common and the longest type of vocalizations produced by little auks[1,27], while other sounds were broad-band, the site soundscape is likely saturated by the trilling calls.

The present dataset is about 100 times longer than the one recorded by[1] and will be valuable for testing the effectiveness of different detection methods[5,6,8] in recognizing the temporal variations in various classes of calls and analyzing their sequence structure[45]—owing to methodological and heavy computational demands, such an undertaking will be published elsewhere. In regard to future studies, we also note that our long-term spectral analysis and inspection of individual audios suggested that because the dominant frequencies were below 10 kHz, a lower sampling frequency of 22 kHz would be sufficient in long-term experiments to save batteries and memory.

## Conclusions

We have demonstrated that sound may offer opportunities to indirectly infer bird-colony activity in remote polar environments. In contrast to the laborious around-the-clock observations made by previous scholars (using telescopes, binoculars, and the naked eye[34,36,37]), or manually counting birds on photos from automated time-lapse photography (as in the Qoororsuaq case), the newer acoustic method is simpler to implement without observer bias and is therefore worthy of further testing.

We identified a nocturnal increase in the sound level at a little auk colony in Greenland under continuous daylight. Noise level can be used as a proxy for colony agitation and the intensity of vocal interactions. This diurnal pattern may be surprising to those who study low/mid-

latitude birds[5]. Still, it recapitulates previous reports of the diurnal activity rhythms seen in the attendance and feeding cycles of the little auk, which peak after midnight during the late nesting period. However, with limited data, contradictory literature claims (about causes of the diel activity rhythm), and lags of various durations depending on the birds' flying time to colony-specific foraging grounds, we could not resolve whether the sound level is proportional to feeding, fledging, or both. It might be that there is no single explanation, as suggested for the mid-latitude bird chorus[16].

Experiments over longer periods and vaster areas (i.e., several colony patches /colonies) should answer these and other questions. They should provide insights into the colony dynamics and the changes with time and across regions in response to external factors. This is important because seabird populations are declining globally[46], while burrow-nesting seabirds, such as the little auk, are some of the most threatened, but challenging to census[12,13]. Seabird colonies in remote and difficult-to-access areas have also been monitored using automatic time-lapse (infrared) cameras, mobile marine radars, and aerial images[12,21,47,48]. Although such imagery can reveal detailed behavior and the causes of events (e.g., predator disturbances), sound provides more-general information and is more widely applicable, most notably for cryptic birds nesting under rocks or in burrows[10,12,13] or birds living in inaccessible cliff locations[14]. Combining imaging and acoustic methods could help to identify limitations and strengths of each method and yield the most efficient monitoring tool[12].

Detailed classification of sounds and their distribution over time may also help to detect the phenology of the breeding season[12]. For example, there may be differences in calls before and after egg laying or after chick hatching, not only with the appearance of chicks as new sound sources, but with the changing nature of inter-pair interactions. Furthermore, routinely recording sound can be convenient for identifying: (1) shifts in colony phenology (depending, for example, on annual food availability or accessibility, weather); and (2) total breeding failure events caused by unusual conditions[49], which are expected to occur more frequently in the Arctic (e.g., Siorapaluk area has been recently affected by landslides).

It is reasonable to suggest that such future efforts will be valuable for understanding the swarming behavior of birds and their synchronization of activities[50], including arrival and departures, large-scale circling flights, and responses to predators. In addition, passive acoustic monitoring of little auk colonies may guide conservation efforts by helping to assess population trends[10,12]. Finally, involving local Inuit communities into jointly co-designed monitoring could facilitate research, while decolonizing science and promoting sustainability.

## Methods
### Observations
The study site was located on a slope facing south near the small village of Siorapaluk (77˚47' N 70˚38' W) in Northwest Greenland (Fig. 1), which was visited in July and August 2022 for various sampling purposes. In this area of Greenland, little auks typically arrive in early May, and lay their eggs in mid-to-late June. The young hatch in mid-to-late July, fledge around mid-August, and leave by the end of August[1,24,34], as apparent in the Qoororsuaq-data collected nearby (described below). Field observations of feeding flocks of little auks suggest that birds from this and/or adjacent breeding colonies forage for food within at least 60 km from their nests ([24,51] authors' pers. obs.), suggesting that the birds leave for extended periods to find food. (In Siorapaluk, chick meals are dominated by *Calanus hyperboreus* and *Calanus glacialis*, further south, i.e., in Thule, and in Svalbard, *Calanus finmarchicus* is also important[52]). We also note that during several summer seasons of oceanographic surveys at sea, we have never noticed any vocalizations from the flocks of little auks outside the colony. Rare observations show that at least some foraging seabirds use sounds at sea[53]. Our personal observation is subjective and needs further verification (i.e., via biologging), but it is based on annual cooperation with local Inuit hunters in the sea since 2015. Hunting and animal observations in open water often required silence, switched-off engine, low wind[54], and often corresponded to submergence

into flocks of little auks commuting between Inglefield Bredning and Siorapaluk.

The sound recorder was hidden between rocks at two locations for approximately 3 days each time. In the first experiment, we aimed to record the background noise near the village ca. 3 km away from the colony (26–29 July 2022); in the second experiment (18–22 August 2022), we recorded sounds directly in the colony during the chick-fledging period. The rationale for the first test was quick access, which might be beneficial in future long-term observations when the power supply and data retrieval become crucial in the setup design.

The sound was collected using a Song Meter Micro (Wildlife Acoustics, Maynard, USA) at a sampling rate of 44.1 kHz (i.e., with a Nyquist frequency of ~22 kHz) with 16-bit resolution. The recorder is compact (~10 × 7 × 3 cm$^3$), lightweight (195 g), weatherproof, and includes a built-in omnidirectional microphone; it runs for about 3 days on three AA alkaline batteries and stores 1 h-long single-channel .wav files on a micro SD card. Self-noise is relatively flat, with a broad 10 dB peak at 16.5 kHz. The microphone has sensitivity of −10 ± 4 dB (0 dB = fs/Pa@1 kHz) which is slightly higher at around 6 kHz. More-detailed technical specifications are available at https://www.wildlifeacoustics.com/products/song-meter-micro.

For a better-constrained interpretation of our acoustic data, we also include an analysis of detailed time-lapse camera records of little auks collected south of Siorapaluk, in the Qoororsuaq colony, near the Pituffic Glacier (76˚16' N 68˚57' W). The number of birds sitting on rocks was counted in a defined count area in August 2013, 2014, and 2016 (Supplementary Fig. 3). The photos were collected at 1 h intervals until all birds had left the colony for the year, and, thus, the data are complete. These data and their analysis correspond to the least subjective, the longest, and the most recent records of the little-auk-colony attendance in Greenland and, therefore, are highly relevant to our work in Siorapaluk.

Both acoustic and camera observations on little auks were noninvasive and, therefore, did not require any institutional/governmental approval or oversight.

### Acoustic data analysis
To analyze the audio data (in total 56 Gb, >144 h), first, we constructed long-term spectrograms (LTSs) and computed the median relative sound intensity, RSI[32,33]. The former corresponds to 3D data (i.e., fast Fourier transform, FFT, with sliding windows), whereas the latter yields 2D data, which are easier to compare with other time series (such as sun elevation and biological rhythms available in the literature). On the one hand, such continuous approach is more robust than call-rate estimation under a condition of overlapping calls saturating detection rates. On the other hand, it can be affected by ambient sounds, like waves, wind or river[12,30], which were however not a concern due to calm conditions of the campaign. To generate LTSs, we followed the procedures of ref. 32. Specifically, LTS was computed with 10 s time resolution using a FFT window size of 1024 samples without overlapping for a frequency range 100–20,000 Hz. This procedure generated spectrograms with the frequency resolution of ~43 Hz.

For a complementary quantitative analysis of the little auk soundscape, we also calculated such common in soundecology biophonic indices as the Acoustic Complexity Index and Bioacoustic Index both of which are known to correlate with the number of calls in a bird colony[13]. To ensure the frequency range of the little auk sounds was included, the frequency limits were set to 0.5–10 kHz; FFT window size was 512 (Supplementary Fig. 1).

The aforementioned analysis was designed to allow an overall, long-term view (i.e., the LTSs were helpful in identifying periods of increased intensity of little-auk calls and sounds, which could be verified by listening).

For recognition of different sound types, their classification, and temporal dynamics, additional quantitative analysis was made to exemplify the composition of the soundscape, as explained below.

An automatic signal detection and clustering analysis based on Hidden Markov Models in Kaleidoscope software produced a training set for the trilling call and wing-flapping sounds. Signal detection parameters were chosen according to the spectral and temporal characteristics of these

sounds (the minimum and maximum frequency range was 6000–7600 Hz, the minimum and maximum detection length were 0.5–2 s, with 0 s inter-syllable gap). For the cluster analysis, default recommended parameters were used (the maximum distance from cluster center 1.0, 512 FFT window size, the maximum number of states 12, the maximum distance to cluster center for building clusters 0.5, and the maximum number of clusters 15). The procedure yielded 931 detections in two clusters. Each detection was then manually reviewed to be labeled accordingly or to be rejected (if unclear or mixed/overlapping). The manually reviewed advanced classifier was then used to re-scan recordings and create new dataset yielding 249 detections (Supplementary Fig. 2). The detector's accuracy was evaluated by a manual review of the entire dataset (accuracy: 97.6%). The model was then applied to the village dataset (July): this resulted in 0 trilling calls and only a few wing-flapping sounds (~once a day), as could be expected due to the distance from the colony. For detecting single calls, the minimum and maximum frequency range was set to 2600–5200 Hz, the minimum and maximum length of detection was 0.08–0.11 s, with 0 s inter-syllable gap, whereas cluster analysis had the same parameters as shown above. This model detected 31,976 signals in 15 clusters. Following a manual review of ~11,000 signals and merging 4 clusters, the accuracy of single-call detections was 99.8%. Application of this model to the village dataset (July), resulted in 513 manually reviewed detections, none of which, however, corresponded to single calls of the little auk.

Finally, to provide a relative comparison of the village and colony datasets, for an automatic signal detection we use 0.5–10 kHz frequency range, 0.1 and 2 s as minimum and maximum length of detection, 0.15 s as maximum inter-syllable gap, and 512 FFT window size. Automatic signal detection yields 347 events per hour for the village data (July), and 2129 events per hour for the colony data (August).

The results are presented in local time (UTC minus 2 h).

## Statistics and reproducibility
The statistical analyses of the data, where applicable, were conducted using the Mann–Kendall test (at the alpha significance level of 0.01). The microphone sampling rate was chosen to completely cover the audible frequency range. The reproducibility of acoustic analyses confirming the biotic nature of the soundscape variation in the colony was ensured using several independent approaches, as detailed in the text and corresponding Methods sections (i.e., aural verification, computation of biophonic indices, and signal detection/classification). To ensure the reproducibility of cycle detection, the sample size (i.e., the data length) had a duration covering at least three cycles, which satisfies the required minimum[15], and which is longer than the most recent observations on little auks[37]. The reproducibility of nocturnal gathering of birds at the end of the season in neighboring and distant locations, has been ensured by literature review[34,35,37], as well as three months of time-lapse-camera data from three different summer seasons (Supplementary Fig. 3; Supplementary Data 1).

## Reporting summary
Further information on research design is available in the Nature Portfolio Reporting Summary linked to this article.

## Data availability
Data are publicly available through Zenodo data repository[55,56]. The source data for Fig. 4 are in Supplementary Data 1; data from other regions are available in Supplementary Data 2 and [34,35,37]. Sun elevation (Fig. 3) was retrieved from https://www.sunearthtools.com/dp/tools/pos_sun.php.

## Code availability
Long-term spectrograms and sound characterization were computed using open Matlab codes by Guan et al.[32] (https://github.com/schonkopf/long-term-spectrogram). Standard functions of Matlab R2018b were used for analysis and plots (https://mathworks.com/products/matlab.html). Previously published data were extracted using open-source Webplotdigitizer: Version 4.6[57]. Audio files were processed in Kaleidoscope 5.5.2 (Wildlife

Acoustics) and inspected in publicly available software Raven Lite 2.0.0 (Cornell Lab of Ornithology).

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

## Acknowledgements

The authors warmly thank colleagues R. Kusaka, S. Sugiyama, and Oshima family for their support during fieldwork, B. Helm, F. Amélineau, and the two anonymous reviewers for their comments on the initial version of this manuscript. This research was supported by an Arctic Challenge for Sustainability research project (ArCS-II; JPMXD1420318865), funded by the Ministry of Education, Culture, Sports, Science and Technology of Japan (MEXT); and JST SPRING, Grant Number JPMJSP2119. Fieldwork in the Qoororsuaq colony was undertaken as part of the Baffin Bay Environmental Study Program 2011–14, funded by the Bureau of Minerals and Petroleum, Greenland Government, and we wish to thank Peter Lyngs for untiring counting of little auks on thousands of time lapse photographs.

## Author contributions

E.A.P. conceived and designed the study, analyzed the results, visualized the data, and wrote the original draft. M.O. undertook fieldwork in the Siorapaluk colony. J.B.T. contributed into writing, reviewing and editing, and provided advice for this study. K.L.J. and A.M. designed and undertook fieldwork in the Qoororsuaq colony and contributed analysis of time-lapse photographs.

## Competing interests

The authors declare no competing interests.

## Ethics

The study was noninvasive and complies with ethics and biosecurity policies of the journal.
