## [Peer Review File · Communications Biology]

Reviewers' comments:

Reviewer #1 (Remarks to the Author):

Summary and general comments:

The authors present a very descriptive study of two new recordings of audio data at a little auk (*Alle alle*) colony in North West Greenland. They recorded a 3-day period in July, a bit far away from the colony, and a second period of 4 days in August, within the colony to record calls. The general messages are that an audio monitoring can be an easy way to monitor activity at a bird colony and that little auks seem to have a 24h cycle (at least during the fledging period, as no rhythmicity was detected in the July recording). The acoustic monitoring of a little auk colony is new, but none of these two main findings are new. Many studies on acoustic monitoring of seabird colonies have been performed before, as well as studies on little auk activity. I was really excited to read a paper on that topic but I am left with many concerns detailed below.

The main criticism is that this study is lacking hypotheses, or at least, in the case of a purely descriptive study, a clear story. As it is, the authors discuss various topics. It goes in different directions and it is not easy to understand why they performed the study and what is the key result they want to give. I am left with the idea that they put recorders without a clear idea of what to do with the data, and other approaches could have been more relevant for some of the points discussed. If the aim was to find the daily activity pattern of little auks, it could have been interesting to have more than two recordings during the breeding season (including incubation for example, even pre-incubation?). It could also have been interesting to couple it with time-lapse photography, another technique to study activity in the colony, or visual observations to confirm that more birds are present when more calls are recorded. Another approach could be to deploy loggers to measure individual foraging patterns and confirm whether they forage more at night or not. If the aim was to describe the different vocalizations of little auks, then another approach recording one/a few birds at a time, and visually assessing the behaviour concomitantly could be more efficient than a colony recording.

The second main criticism is that the authors are not critical enough regarding the activity pattern they observed. The daily pattern was only observed during one of the two recordings, but in the abstract it seems that it is happening during the whole breeding season. Also, the authors cite another study on little auks (ref 23, Wojczulanis-Jakubas et al. 2020) and claim that they found a 24h-hour cycle, but this is not the result of this study. In reference 23, they find an average cycle of 23.2h [range from 19.9 hr to 30.8 hr]. This is incorrectly reported as a 24h cycle in the present manuscript. The fact that the average cycle is not exactly 24h is an important result that cannot be underestimated when studying cyclicity in polar environments. Figure 4 makes the result of the 24h-cycle pretty convincing but it is not correct to not describe studies that do not find a cycle, or a cycle different than 24h (ref 23).

Another important point is that the second recording period happens when chicks are fledging. It is well known that chicks mostly fledge at night, and it could be that the activity is higher at night during that period because of the fledging activity, and the presence of fathers encouraging the departure of the chicks. This is just mentioned in the supplementary information but should be discussed in the main text, especially because no diurnal pattern has been found in July and this could be one possible explanation.

The fact that little auks have various vocalizations, described in the introduction and in the discussion is to me the main interest of this new dataset, but is not described here. I strongly support the authors to publish the study they describe in the discussion on vocalization types "such an undertaking will be published elsewhere" L146, and I am looking forward to reading this study. As the manuscript is at the moment, I do not recommend it for publication in *Communications Biology*.

I acknowledge the effort of the authors to search in sometimes old literature for studies on little auk calls. It could have been interesting to broaden the search to other Arctic seabirds (see for example work of Nicholas Per Huffeldt), or even Antarctic seabirds that are facing the same conditions.

Some other references mentioning a diurnal cycle in polar seabirds (non-exhaustive list):
Klages, Norbert. "Food and feeding ecology of emperor penguins in the eastern Weddell Sea." *Polar Biology* 9 (1989): 385-390
Paulin, C. D., and P. M. Sagar. "A diurnal rhythm of activity by the Adélie Penguin." *Notornis* 24 (1977): 158-160.
Huffeldt, Nicholas Per, and Flemming R. Merkel. "Sex-specific, inverted rhythms of breeding-site attendance in an Arctic seabird." *Biology letters* 12.9 (2016): 20160289.

Finally, another small general comment is about the assumption of the authors that little auk activity is low in the presence of glaucous gulls. In big little auk colonies, in the presence of gulls, birds usually fly in circles and emit calls. It is not obvious to me that the colony would be more silent in the presence of gulls.

Specific comments:

Abstract:

The first sentence assumes the reader is familiar with the answer. It could have been great to detail this assumption in the introduction.

See general comment on the presentation of the 24h cycle as the general pattern in little auks. You did not observe it during the first recording period but you don't mention that in the abstract (it seems that it happens all the time).

Introduction:

L10-11: "how birds time their behaviour is a fundamental question" could you detail why?

Results:

L60: "some high amplitude transients": what does it mean? Is there some biological meaning?

Discussion:

L159: "contradictory literature claims": you could give more details on that in your manuscript

Methods:

L187-188 and L197-198: you give two different timings for the fledging period. Which one is correct?

L191-193: this sentence comes a bit out of nowhere. If you leave it, you need to give more details about the frequency of observations, the location, etc. Were you in conditions where you could detect vocalizations at sea (e.g. far away from a ship's engine noise, low wind conditions...)?

- You don't explain how you extracted the data from the previous studies to make fig. 4.

References:

Number 17: author names don't have a capital letter

Figures, tables, supplementary materials:

Table 1: add author name, location of the study

Fig. 2: panel a and c, you could add the time on the x-axis in addition to the day.

Fig. 3: same as fig. 2.

Fig. 4, legend: "number of Calanus/surface 10 m³" is it the number of Calanus per m³ of water (abundance)? If yes, it is a volume and not a surface. The y-axis for this variable is also unclear, I suggest "abundance of Calanus (10⁻¹ ind/m³)"

Is there a way to present error bars in this figure? For your study and for some of the other studies there are data for more than one day.

Please add a visual legend for bold black lines too (and not just in the top panel).

Supp. Fig. 1 and supplementary note 2: your observations are very subjective. What do you mean by "without any high-intensity behavioural response from the little auks"? On your picture, they are flying. Isn't it the response from the presence of gulls and raven?

Supplementary note 1: this should be in the main text. Please cite the author name (and not just the reference number) to make the text easier to follow (as you did in the first sentence of this note).

Reviewer #2 (Remarks to the Author):

The manuscript entitled "Bird calls under continuous daylight: Arctic colony activity revealed by sound" submitted to *Communication Biology* present an interesting study on the colony activity of little auks in Greenland using passive acoustic monitoring. The manuscript addresses the important question of how seabirds adapt to the continuous daylight environment during the Arctic summer. The results have immediate relevance to researchers interested in Arctic ecology, and the study provides valuable insight into the behavior of little auks in their natural habitat.

However, I believe there is still much room for improvement in the manuscript. Firstly, the authors only briefly analyze the diurnal variation of soundscapes and do not provide any detailed investigation on how the acoustic behavior of little auks shape soundscape characteristics. Moreover, the authors use uncalibrated sound intensity as an indicator of little auk colony activity. The use of sound intensity in assessing animal behaviors may be easily affected by environmental and anthropogenic noise, thus it is crucial to properly filter audio recordings or to investigate the relationship between sound intensities and little auk abundance for ensuring the reliability of acoustic analysis. Additionally, the authors provide only three examples of little auk calls (wind flapping should not be considered as a vocalization) and do not perform any call detection or quantitative repertoire analysis. These are fundamental techniques in passive acoustic monitoring of wildlife and should be included to support the assessment of little auk colony activity.

Finally, the authors claim that the strong activity pattern observed parallels the foraging behavior of little auks on vertically migrating zooplankton. However, the study lacks direct observations of feeding behavior or in-situ data associated with zooplankton abundance. Therefore, the conclusion is not entirely convincing and requires further investigation.

Overall, I suggest that the authors address the weaknesses highlighted above and incorporate call detection and quantitative repertoire analysis to provide a more comprehensive understanding of the little auk behavior. Additionally, the authors should provide data of feeding behaviors to support their conclusions.

Specific comments:

Abstract:

1. Change "noise cycle" to "acoustic activity cycle".
2. As there is no direct observation of vertically migrating zooplankton, it is recommended to soften the claim linking the strong activity pattern to foraging behavior.
3. Instead of describing PAM as elegant, it is suggested to acknowledge it as a complementary tool to manual observations. Also, briefly highlight the future application of this technique for studying wildlife behavior in the Arctic.

Introduction:

1. The introduction could start by asking whether polar seabirds have diel behaviors in continuous daylight, followed by highlighting the advantage of passive acoustic monitoring for studying such behaviors.
2. Line 7: It is important to clarify that although there are many studies on continuous acoustic monitoring of bird colonies, there is a lack of such studies in polar regions.

3. Line 16: It would be helpful to provide more explanation on why little auk is an ecological engineer in Arctic marine and terrestrial ecosystems, especially regarding how they structure these ecosystems.
4. Line 30: Instead of describing the lack of recordings and quantitative studies, the authors could clarify the key question addressed in this study.
5. Line 40: The final part of introduction appears to repeat the concluding remarks. It is suggested to remove this part or revise it to emphasize the primary question addressed in this study.

Results:

1. The results section could begin by briefly introducing the recording sites and what the authors are comparing in this study.
2. Line 46: Remove "i.e., fast Fourier transform, FFT, with sliding windows".
3. Line 47: Change "the sound" to "soundscape" to more accurately reflect the focus of the analysis.
4. Clarify the purpose of trying to identify little auk-related sounds from the long-term spectrograms, and remove the description of spectral peaks associated with microphone sensitivity and ambient noise as they are not relevant to little auk acoustic behaviors.
5. Lines 60-65: Justify the use of broadband SPL as an indicator of bird audio activity.
6. Line 71: Lack of supporting statistical analysis for the daily decrease in amplitudes.
7. Line 72: Provide a clear definition of day and night under continuous daylight.

Discussion

1. Line 95: Clarify what is meant by "global rhythmicity". If little auk make sounds during movement, two peaks should be expected for leaving and returning, but the absence of such a pattern needs to be explained.
2. Line 120: Address whether any statistical analysis supports the relationship between bird numbers and sound levels.
3. Line 130: Clarify the meaning of "global spectral analysis".
4. Line 143-146: Instead of simply mentioning that other results will be published elsewhere, provide suggestions for future data collection and analysis, possibly with complementary methods. This will enhance the manuscript's value for researchers working in Arctic ecology.

Conclusions:

1. Line 148: I suggest revising this statement to be more cautious and accurate. While the analysis presented in the manuscript does not provide direct evidence of a link between sound and little auk behavior, the audio data may offer opportunities to indirectly infer colony activity and foraging behavior (still need to be added by detecting foraging sounds).
2. Line 166: Add "and satellite images".
3. Line 178: It would be beneficial to provide more specific suggestions for how passive acoustic monitoring of little auk can guide conservation actions, such as identifying key foraging areas or assessing population trends. Additionally, the authors could discuss how involving local communities, such as the Inuit, in the monitoring process could promote sustainable development and support conservation efforts.

Methods:

1. Line 185: Remove "to be published elsewhere"
2. Line 188: Clarify the statement about all little auks leaving on September 1st to avoid any confusion or overgeneralization.
3. Line 206: Provide the sensitivity information of the recording system.
4. More details on the soundscape analysis could be provided in the manuscript, such as the procedures to generate long-term spectrograms and how the LTSs were used in identifying little auk calls.

Table 1:

1. Includes the authors' names in a separate column, which makes it easier to quickly identify the

researchers associated with each study. Additionally, add a short description of the primary finding and survey methods will make the table more informative.

Figure 2:

1. Change the y-axis label to "Frequency (kHz)" to avoid showing $\times 10^4$. Also, adjust the x-axis limit to exclude periods without observation.
2. Convert the power spectral density (PSD) plot to a probability distribution to enhance readability.
3. Consider calibrating the recording sensitivity, such as the value listed on the recorder specification, to display the actual dB scale instead of relative values.
4. Focus on frequencies below 10 kHz and expand the low-frequency range. Use arrows or boxes to highlight the relevant features of the spectrogram.

Figure 3:

1. Where is the data recorded from the village site? Exclude periods without observation.
2. You can't put SPL as the y-axis label if you didn't calibrate the recording sensitivity.

Figure 4.

1. The 1974 data may not be too old for supplement the current observation result. The climate condition and behavior of little auk may be very different.
2. Remove redundant panel, consider add more statistical analysis to support your statement.

Reviewer #3 (Remarks to the Author):

The study seems to examine a diel rhythm of vocal activity of a polar seabird. Diel rhythmicity of various activities both of animals and humans in conditions of continuous daylight remains an intriguing issue. Still not many species has been properly examined in this context, thus mechanisms and taxonomical patterns are not fully recognized.

The study species is truly polar animal (endemic for the Arctic), and so ideal for examining the topic of daily activity patterns in the conditions of polar day. As a matter of fact such an issue has been already considered (specifically colony attendance patterns and foraging activity has been examined in respect to the time of the day). However, the issue has been examined locally and so it is still not clear how universal are the findings then there is still a vast area of various research questions that could be asked.

Unfortunately the present study has a serious flaw, and I believe that it cannot be fixed based on the presently used data. First the aim of the study is very much unclear, then whatever the aim is the design of the study is not really adequate. Below I elaborate on this and provide a list of more specific comments on the manuscript content. The latter are also sometimes quite serious (being linked to the major concerns).

Major concerns:

1. Unclear aim of the study and so the narration: I am very much confused is the study about diel rhythmicity of the little auk in given site and period? or is it rather about an approach to examine the issue, i.e. with passive acoustic recording. In any case it is also unclear is the study about just birds vocal activity or their colony attendance? Then, at some point a vocal repertoire is mentioned as a study aim. The whole manuscript sounds like a mixture of everything but then applied methodology is not really adequate to any of the issues (see more below). Whatever the aim would be, also its background and interpretation should be elaborated (considering broader ecological/evolutionary context, relevant findings on other species, etc).

2. Design of the study: Since the aim of the study is unclear it is also hard to evaluate correctness of the applied methodology. However, whatever the aim would be the study seems to be not properly measuring/sampling the reality.

For studying diel rhythmicity (no matter whether that would be vocal activity or colony attendance, or whatever) a longer period of time should be considered. Three days of acoustic recording at the end of the chick rearing period represents just an anecdotal data. Selected three days could be very specific, for example due to weather conditions. Then the end of the season (around fledging) has its specificity, especially in the little auk (it is well known that little auk young depart the colony during night hours). Whatever happens during that time is characteristic to that time and cannot be extrapolated to other periods, neither compared with other data, if those have been collected in different breeding stages (as it is the case on Fig 4).

For evaluating applied methodology, not only control in the non-colony spot (here a village) should be considered but also the control at the colony area (for example, video recording of the birds presence). Again, no matter what is actually considered here - just vocal activity or colony attendance, one has to be controlled for the other. This is because the birds can be in the colony but they may remain quiet (and this is what really happens in the field, although that is only the case when birds are in the colony in low numbers, but this is the part of reality). Besides, more recording spots should be considered as setting the devices in a single spot may give information on this random spot only. If one wants to measure the phenomenon at the colony level, different colony areas should be examined/sampled, to account for potentially biasing local effect.

For vocal repertoire, again one has to keep in mind the breeding stage, as the repertoire may change during the breeding season, and especially fledging time may have its specificity. Then, just pointing out few calls (out of many more in the little auk) without any context (e.g. is the signal emitted in a social interaction, at given breeding circumstances, who is the producer (male/female/chick)) is not really informative. Then, other sounds in the colony (like a colony chorus, wing beats) are not really a part of vocal repertoire.

Other concerns (of various gravity, presented in "chronological" order):

Lines 23-36: Is this really relevant in the study?

Lines 37: Not clear what gaps in knowledge this is to fill up (see major comment)

Line 41-42 "(2) demonstrate that the ambient sound is an efficient proxy to colony activity" – with no proper control for the birds colony attendance and a single recording spot this cannot be demonstrated.

Line 43: document vocalization repertoire of the species – you present very limited number of calls and without any context it is not really informative (see major comment)

Line 45: The 56Gb (>144 h) - may sounds impressive but in the context of the study question (whatever it would be) it may not be enough. Besides such a general information is misleading here, as this is not 6 days of continuous recording in given spot, but 3 days in two spots (and that makes a big difference).

Lines 45-52: It is unclear what analysis and why was performed here. I do understand that methods are presented at the bottom of the manuscript (journal style) but if so, the results should be

presented in a way that one does not have to go to methods to the end of the text to understand what is before, in the results section.

Lines 48-51: Unclear what it is about.

Lines 54 "arbitrarily chosen audio files" - why arbitrary? This is not the best approach to examine vocal repertoire of a species.

Lines 56: what is continuous chorus of the colony? – overlapping (various) signals of many individuals? But this cannot really be considered a part of the little auk vocal repertoire

Lines 56: wing flapping – is not a part of the little auk vocal repertoire, it may be a part of the acoustic landscape of the little auk colony but without any context is hard to establish its (biological) meaning.

Lines 51-52 "(suggesting that the vocalizations were insufficient to saturate the ambient noise because the distance involved)" - unclear

Figure 3: median-filtered time-series calculated – unclear what does it mean?

Lines 67-68: It is really hard to see this pattern. I mean, one can see the sinusoid but cannot really point out the exact hours.

Lines 69-70: unclear, hard to see the exact duration of the lags, also unclear what "(mean and standard deviation values for six peak{trough pairs)" means

Line 71: and what is the reason, why it is important – with three days recording it is just a change not really the pattern.

Figure 4. The breeding period considered here and in the other publications is very different and cannot be really compared that easily. Besides it is unclear how these literature values were extracted (e.g. how number of feedings in given hour were calculated? actually the number of feedings (per nest/colony??) sounds complicated to use here).

Lines 89-92: Unclear

Table 1: Counting gulls (any predator) in the colony is tricky as they adjust their presence to the presence of the prey

Lines 101 "slightly" – rather considerably!

Lines 101-106: Unclear how the lag was calculated, and presented (one is supposed to guess it/calculate on their own from the fig 3?)

Lines 111-113: Hard to see the logic behind. You cannot accept one of the hypotheses (that requires a separate testing) just excluding the others (which you actually have not tested properly).

Lines 191-192: It does not mean anything. You could not hear it simply because of being too far/at different time/etc. Not hearing the birds vocalizing at sea (while not really measuring it properly) you can not conclude they do not vocalize while at sea.

Lines 201-203: What format were you recording?

Lines 210-211 – unclear why you constructed the two types of the spectrograms and what is the difference between them.

Lines 213-214: Unclear

Submitted: 9 Dec 2022

Reviews: 14 Mar 2023

Dear Editors and Reviewers,

Thank you very much for investing time and effort into evaluation of our manuscript. Our detailed point-by-point responses are provided below (shown in **bold**; modifications are highlighted in the manuscript; to ease responding, we took liberty to split some paragraphs). We believe the quality of the paper has been improved by revisions and kindly ask to read them and our responses.

Yours sincerely,
Evgeny Podolskiy
and co-authors

Reviewers' comments:

Reviewer #1 (Remarks to the Author):

Summary and general comments:

The authors present a very descriptive study of two new recordings of audio data at a little auk (*Alle alle*) colony in North West Greenland. They recorded a 3-day period in July, a bit far away from the colony, and a second period of 4 days in August, within the colony to record calls.

In contrast to previous studies, this is the first paper with non-invasive observer-independent data revealing arctic-bird activity. Thus, “very descriptive” is misleading.

The general messages are that an audio monitoring can be an easy way to monitor activity at a bird colony and that little auks seem to have a 24h cycle (at least during the fledging period, as no rhythmicity was detected in the July recording). The acoustic monitoring of a little auk colony is new, but none of these two main findings are new.

We humbly note that none of these messages were brought forward in the Abstract as “two main findings”. Instead, we found previously unknown acoustic excitement after midnight. The existence of acoustic cycle in little auk should not be confused with messages putting it into context and clearly citing previous literature (on acoustic monitoring in other birds and observer’s counts).

Many studies on acoustic monitoring of seabird colonies have been performed before, as well as studies on little auk activity. I was really excited to read a paper on that topic but I am left with many concerns detailed below.

For the Arctic, we found no papers. It will also be an overstatement to suppose that there have been many studies on little auk activity (e.g., there are just 3 papers documenting a diel cycle, with no agreement on its cause).

The main criticism is that this study is lacking hypotheses, or at least, in the case of a purely descriptive study, a clear story. As it is, the authors discuss various topics. It goes in different directions and it is not easy to understand why they performed the study and what is the key result they want to give.

We respectfully disagree with this judgment. The addressed scientific question was mentioned in the Introduction, where we hypothesized the existence of a circadian-like rhythm in the sound activity. Therefore, the main topic at the center of the story, is the oscillating soundscape at a little auk colony. In Discussion, we compare it to the existing knowledge on birds activity cycle. The study is performed to show the existence of this previously unknown acoustic phenomenon, which is the key result. We hope revisions to the text and title have improved the manuscript.

I am left with the idea that they put recorders without a clear idea of what to do with the data, and other approaches could have been more relevant for some of the points discussed.

Suggestion that an acoustician and ornithologists from Japan are coming to the most northern settlement in Greenland to “put recorders without a clear idea of what to do with the data” is upsetting, but we understand the Referee did not mean to humiliate us. We clearly stated (1) that almost nothing is known about acoustics of a little- auk colony, and (2) explained why locations for each survey were tried. It is also misleading to state that “other approaches could have been more relevant”, because there is no silver bullet for studying seabirds with one approach (as explained below).

If the aim was to find the daily activity pattern of little auks, it could have been interesting to have more than two recordings during the breeding season (including incubation for example, even pre-incubation?).

Having significantly different breeding stages is wonderful, however, one first needs to confirm that acoustic recordings are usable. As it was not guaranteed a-priori, we had to select the period which would be possible to relate to the literature on diel activity of little auks. And the highest number of studies is available for chick-rearing period.

It could also have been interesting to couple it with time-lapse photography, another technique to study activity in the colony, or visual observations to confirm that more birds are present when more calls are recorded.

In the initial manuscript, we acknowledged that cameras are of a limited use for little auks nesting under rocks. Furthermore, while visual observations on attendance have been already available in the literature (and we had no intention to question them), they remain subjective, possibly invasive, and, most importantly, not feasible for a long-term monitoring.

Another approach could be to deploy loggers to measure individual foraging patterns and confirm whether they forage more at night or not.

Individual foraging patterns of little auks (or their predators) could be great studies, which appear to be meaningful to pursue considering non-conclusive interpretation of the diel cycle. Our study, however, aimed to detect the existence of a circadian-like rhythm in the colony soundscape in a non-invasive way.

If the aim was to describe the different vocalizations of little auks, then another approach recording one/a few birds at a time, and visually assessing the behaviour concomitantly could be more efficient than a colony recording.

As explained below, fine descriptions of vocalizations were not the main aim, especially since this was attempted previously.

The second main criticism is that the authors are not critical enough regarding the activity pattern they observed. The daily pattern was only observed during one of the two recordings, but in the abstract it seems that it is happening during the whole breeding season.

This detail was overlooked in the Abstract, but crystal clear in the text. We clarified the time, so there should be no ambiguity now. Nevertheless, there was no overstatement since personal accounts of noisy night-hours in the early breeding season were cited.

Also, the authors cite another study on little auks (ref 23, Wojczulanis-Jakubas et al. 2020) and claim that they found a 24h-hour cycle, but this is not the result of this study. In reference 23, they find an average cycle of 23.2h [range from 19.9 hr to 30.8 hr]. This is incorrectly reported as a 24h cycle in the present manuscript.

Nowhere in the paper we explicitly “report/claim that they found a 24h-hour cycle”. In this regard, ref. 23 appears only implicitly in Table 1. Its caption was “Possible drivers of a 24h cycle...” To avoid misunderstanding, we replaced “24h” with “diel”.

The fact that the average cycle is not exactly 24h is an important result that cannot be underestimated when studying cyclically in polar environments. Figure 4 makes the result of the 24h-cycle pretty convincing but it is not correct to not describe studies that do not find a cycle, or a cycle different than 24h (ref 23).

Signal-processing theory does not allow to establish “that the average cycle is not exactly 24h” from 48h of data in ref.23. With such short data-length (~2 cycles), one neither can achieve high-frequency resolution (Zielinski et al., PLOS One, 2014), nor compute periods

longer than 1 cycle (Cheveigné & Kawahara, JASA, 2002). We mention their finding, but hesitate to overstate its importance. We have also described “studies that do not find a cycle” in other polar seabirds.

Another important point is that the second recording period happens when chicks are fledging. It is well known that chicks mostly fledge at night, and it could be that the activity is higher at night during that period because of the fledging activity, and the presence of fathers encouraging the departure of the chicks. This is just mentioned in the supplementary information but should be discussed in the main text, especially because no diurnal pattern has been found in July and this could be one possible explanation.

Thank you for this idea! We have mentioned this observation, but could not find quantitative support for it. Also, please note, that in the beginning of Results, it was stated that no-pattern in July stemmed from non-biological reasons.

The fact that little auks have various vocalizations, described in the introduction and in the discussion is to me the main interest of this new dataset, but is not described here. I strongly support the authors to publish the study they describe in the discussion on vocalization types “such an undertaking will be published elsewhere” L146, and I am looking forward to reading this study. As the manuscript is at the moment, I do not recommend it for publication in Communications Biology.

We have added more quantitative analysis of vocalization types. Yet, we note, that focusing on the trees (discrete sounds), instead of the forest (soundscape) was not the main scope of our study. The former requires a full-length paper, more suitable for specialized journals; e.g., only 1.5h of data corresponded to at least 11 figures in the previous study (Ferdinand, 1969). This is not an excuse, but opinion based on experience: for poorly documented sounds, ~2 minutes / ~45 minutes / or ~17 hours of audio corresponds to separate papers [Podolskiy, GRL, 2020; Podolskiy et al., JASA, 2021; Podolskiy and Sugiyama, JGR, 2020], while we are talking about more than 100 h of data with overlapping chorus of birds.

I acknowledge the effort of the authors to search in sometimes old literature for studies on little auk calls. It could have been interesting to broaden the search to other Arctic seabirds (see for example work of Nicholas Per Huffeldt), or even Antarctic seabirds that are facing the same conditions. Some other references mentioning a diurnal cycle in polar seabirds (non-exhaustive list): Klages, Norbert. "Food and feeding ecology of emperor penguins in the eastern Weddell Sea." *Polar Biology* 9 (1989): 385-390; Paulin, C. D., and P. M. Sagar. "A diurnal rhythm of activity by the Adélie Penguin." *Notornis* 24 (1977): 158-160; Huffeldt, Nicholas Per, and Flemming R. Merkel. "Sex-specific, inverted rhythms of breeding-site attendance in an Arctic seabird." *Biology letters* 12.9 (2016): 20160289.

Thank you very much for these suggestions, we have included them all.

Finally, another small general comment is about the assumption of the authors that little auk activity is low in the presence of glaucous gulls. In big little auk colonies, in the presence of gulls, birds usually fly in circles and emit calls. It is not obvious to me that the colony would be more silent in the presence of gulls.

Nowhere in the paper we made such assumption explicitly. The presence of gulls is mentioned as a previously suggested factor affecting the attendance. We went through the text and tried to make sure that this is clear.

Specific comments:

Abstract:

The first sentence assumes the reader is familiar with the answer. It could have been great to detail this assumption in the introduction.

This is a good point. We detailed the answer as advised.

See general comment on the presentation of the 24h cycle as the general pattern in little auks. You did not observe it during the first recording period but you don't mention that in the abstract (it seems that it happens all the time).

We have clarified that the cycle was detected in August.

Introduction:

L10-11: "how birds time their behaviour is a fundamental question" could you detail why?

We have added that "because it is relevant to all aspects of avian biology, including social interaction, reproduction, migration, orientation, and vocalization".

Results:

L60: "some high amplitude transients": what does it mean? Is there some biological meaning?

This is a common language for describing signals — without a reference to source mechanisms — when describing non-stationary, transient signals. To clarify, we specified that this refers to signals.

Discussion:

L159: "contradictory literature claims": you could give more details on that in your manuscript

We have specified that this referred to causes of the diel activity rhythm (more details have been also provided to the Discussion, Table 1, and Suppl. Note 1).

Methods:

L187-188 and L197-198: you give two different timings for the fledging period. Which one is correct?

Day-scale precision of the first instance (L187-188) was criticized by Reviewer #2. We removed such over-optimistic precision to avoid misunderstanding (for example, 12 Aug can be termed as "pre-fledging" (Frandsen et al., 2014), while 14 Aug can have "nearly fledged nestlings" (Roby et al., 1981), and etc).

L191-193: this sentence comes a bit out of nowhere. If you leave it, you need to give more details about the frequency of observations, the location, etc. Were you in conditions where you could detect vocalizations at sea (e.g. far away from a ship's engine noise, low wind conditions...)?

We work with local Inuit hunters in the sea since 2015. Hunting and our observations on marine mammals in open water required silence, switched-off engine, low wind [Podolskiy and Sugiyama, JGR, 2020], and often corresponded to submergence into flocks of little auks commuting between Inglefield Bredning and Siorapaluk. Indeed, we work towards eliminating subjective experience from animal observations, yet the contrast between the colony and the sea is so stunning that we prefer to keep it.

- You don't explain how you extracted the data from the previous studies to make fig. 4.

We digitized it using an open-source software (now cited).

References: Number 17: author names don't have a capital letter

Corrected

Figures, tables, supplementary materials:

Table 1: add author name, location of the study

This was provided in the text, and now repeated in Table.

Fig. 2: panel a and c, you could add the time on the x-axis in addition to the day.

Minor 2-h x-ticks have been added.

Fig. 3: same as fig. 2.

Hourly x-ticks were added.

Fig. 4, legend: "number of Calanus/surface 10 m³" is it the number of Calanus per m³ of water (abundance)? If yes, it is a volume and not a surface. The y-axis for this variable is also unclear, I suggest "abundance of Calanus (10⁻¹ ind/m³)"

Thank you for this advice. The y-axis was updated. The legend was correct ("surface 10 m³" referred to 10 cubic meters of water taken along the surface to count the numbers of calanus).

Is there a way to present error bars in this figure? For your study and for some of the other studies there are data for more than one day.

For comparable stage, confidence limits were available in [Wojczulanis-Jakubas et al. 2020] and have been included.

Please add a visual legend for bold black lines too (and not just in the top panel).

A legend has been added.

Supp. Fig. 1 and supplementary note 2: your observations are very subjective. What do you mean by "without any high-intensity behavioural response from the little auks"? On your picture, they are flying. Isn't it the response from the presence of gulls and raven?

During our visits, we never saw an empty sky without a cloud of flying birds, so it is difficult to use this criteria as a response. Unfortunately, non-subjective (objective) observations on little-auk predators do not exist (e.g., even previously published gull census efforts are referred to by the Reviewer #3 as "tricky").

Supplementary note 1: this should be in the main text. Please cite the author name (and not just the reference number) to make the text easier to follow (as you did in the first sentence of this note).

The information of this note is condensed into Table 1 and presents important, but pretty granular details, which discuss incoherence of previous literature. Indeed it justifies a need for further studies, but somewhat deviates from the discussion of our own results. We appeal to the Editor to recommend whether lengthening the main text with this note is preferable. We tried to adjust citation formatting, while we have to follow the citation formatting guidelines of the journal which uses numeric citations.

Reviewer #2 (Remarks to the Author):

The manuscript entitled "Bird calls under continuous daylight: Arctic colony activity revealed by sound" submitted to Communication Biology present an interesting study on the colony activity of little auks in Greenland using passive acoustic monitoring. The manuscript addresses the important question of how seabirds adapt to the continuous daylight environment during the Arctic summer. The results have immediate relevance to researchers interested in Arctic ecology, and the study provides valuable insight into the behavior of little auks in their natural habitat.

Thank you for this summary.

However, I believe there is still much room for improvement in the manuscript. Firstly, the authors only briefly analyze the diurnal variation of soundscapes and do not provide any detailed investigation on how the acoustic behavior of little auks shape soundscape characteristics.

Diel variation of soundscape is the key finding of the paper based on commonly used acoustic metric. How discrete sounds contribute to it is a separate research question, which was partially addressed by Ferdinand (1969). Nevertheless, to address this concern, we have included complementary quantitative analysis of sound types.

Moreover, the authors use uncalibrated sound intensity as an indicator of little auk colony activity. The use of sound intensity in assessing animal behaviors may be easily affected by environmental and anthropogenic noise, thus it is crucial to properly filter audio recordings or to investigate the relationship between sound intensities and little auk abundance for ensuring the reliability of acoustic analysis.

We used the most commonly employed recorder (Perez-Granados and Traba, Ibis, 2021). Moreover, relative energy has been commonly employed in ornithology (e.g., Orben et al., PeerJ, 2019). We agree and acknowledged that, in general, there might be other factors affecting signal-to-noise ratio. Yet, as we are not estimating bird density, this concern becomes minor, especially in the context of Fig. 4, showing the principal existence of a diel cycle detected by independent methods by various scholars. Furthermore, we computed other acoustic metrics (indices and detections) in support of sound-intensity interpretations.

Additionally, the authors provide only three examples of little auk calls (wind flapping should not be considered as a vocalization) and do not perform any call detection or quantitative repertoire analysis. These are fundamental techniques in passive acoustic monitoring of wildlife and should be included to support the assessment of little auk colony activity.

We have provided call detection and quantitative repertoire analysis. However, in the initial manuscript, we stressed that particular calls were provided only as examples of a soundscape. Continuous noise analysis is not less fundamental than other techniques in bioacoustics; especially, then classical call detection becomes unreliable for too many overlapping calls of the colony leading to saturation of call rates (Orben et al., 2019). Wing flapping was discussed as part of soundscape and we made sure that there is no confusion in this regard.

Finally, the authors claim that the strong activity pattern observed parallels the foraging behavior of little auks on vertically migrating zooplankton. However, the study lacks direct observations of feeding behavior or in-situ data associated with zooplankton abundance. Therefore, the conclusion is not entirely convincing and requires further investigation.

Our interpretation is rooted in the most relevant previously published literature on little auks, with the corresponding data on direct observations on feeding and zooplankton explicitly shown in Fig. 4. Furthermore, diel vertical migration of zooplankton in the Arctic is an established scientific fact; for example, bowhead whales foraging on the same copepods in

the region, also follow the migrating zooplankton (more citations were added). Following another comment by the Referee, in Abstract we wrote that this is likely, and clearly stated in conclusions that our study identified the need for further investigation.

Overall, I suggest that the authors address the weaknesses highlighted above and incorporate call detection and quantitative repertoire analysis to provide a more comprehensive understanding of the little auk behavior. Additionally, the authors should provide data of feeding behaviors to support their conclusions.

We have tried to address the remarks as explained above and below. As suggested, call detection and quantitative repertoire analysis have been incorporated. As we mentioned previously, feeding is presented as a hypothesis, which was adopted from previous studies and consistent with behaviour of other copepod predators in the area. In 2024, we plan to retrieve an Acoustic Zooplankton Fish Profiler deployed in the area, and hope to see further support for the behaviour of little auk.

Specific comments: Abstract:

1. Change "noise cycle" to "acoustic activity cycle".

Changed.

2. As there is no direct observation of vertically migrating zooplankton, it is recommended to soften the claim linking the strong activity pattern to foraging behavior.

We softened with "likely".

3. Instead of describing PAM as elegant, it is suggested to acknowledge it as a complementary tool to manual observations. Also, briefly highlight the future application of this technique for studying wildlife behavior in the Arctic.

It might be misleading to label acoustics as complementary, when manual observations are subjective, invasive, and not feasible over prolonged periods of time in remote areas. Thus, we rewrote as a promising tool in remote Arctic areas.

Introduction: 1. The introduction could start by asking whether polar seabirds have diel behaviors in continuous daylight, followed by highlighting the advantage of passive acoustic monitoring for studying such behaviors.

Such starting line was adapted.

2. Line 7: It is important to clarify that although there are many studies on continuous acoustic monitoring of bird colonies, there is a lack of such studies in polar regions.

Clarified.

3. Line 16: It would be helpful to provide more explanation on why little auk is an ecological engineer in Arctic marine and terrestrial ecosystems, especially regarding how they structure these ecosystems.

More explanation has been provided from the corresponding citation.

4. Line 30: Instead of describing the lack of recordings and quantitative studies, the authors could clarify the key question addressed in this study.

This has been addressed by following the first advice to the Introduction.

5. Line 40: The final part of introduction appears to repeat the concluding remarks. It is suggested to remove this part or revise it to emphasize the primary question addressed in this study.

We have to follow the mandatory formatting guidelines of the journal (“The final paragraph should be a brief summary of the major results and conclusions.”)

Results:

1. The results section could begin by briefly introducing the recording sites and what the authors are comparing in this study.

More information of this kind was added.

2. Line 46: Remove "i.e., fast Fourier transform, FFT, with sliding windows".

Moved to Methods.

3. Line 47: Change "the sound" to "soundscape" to more accurately reflect the focus of the analysis.

Changed.

4. Clarify the purpose of trying to identify little auk-related sounds from the long-term spectrograms, and remove the description of spectral peaks associated with microphone sensitivity and ambient noise as they are not relevant to little auk acoustic behaviors.

The purpose was clarified while addressing the first suggestion to the section. Removing the description of instrumental peaks removes the opportunity to say that they were not relevant to little auk, and may cause misunderstanding. Also, only in the next paragraph we establish whether ambient noise is relevant or not.

5. Lines 60-65: Justify the use of broadband SPL as an indicator of bird audio activity.

References and details have been included.

6. Line 71: Lack of supporting statistical analysis for the daily decrease in amplitudes.

Statistical analysis has been provided to the text and Fig. 3. Non-stationary analysis with Mann-Kendall test rejects the null hypothesis of trend absence at the alpha significance level of 0.01; p-value <0.01.

7. Line 72: Provide a clear definition of day and night under continuous daylight.

For the context of the phrase about 2 files, their exact record hour has been shown.

Discussion

1. Line 95: Clarify what is meant by "global rhythmicity". If little auk make sounds during movement, two peaks should be expected for leaving and returning, but the absence of such a pattern needs to be explained.

We replaced “global” with “biological” and elaborated on the suggested speculation.

2. Line 120: Address whether any statistical analysis supports the relationship between bird numbers and sound levels.

In the beginning of the section, we carefully acknowledged that straightforward comparison of sound levels with direct observations in other areas might be misleading.

3. Line 130: Clarify the meaning of "global spectral analysis".

We replaced "global" with "longterm".

4. Line 143-146: Instead of simply mentioning that other results will be published elsewhere, provide suggestions for future data collection and analysis, possibly with complementary methods. This will enhance the manuscript's value for researchers working in Arctic ecology.

Suggestions for future with complementary methods were provided in Conclusions.

Conclusions:

1. Line 148: I suggest revising this statement to be more cautious and accurate. While the analysis presented in the manuscript does not provide direct evidence of a link between sound and little auk behavior, the audio data may offer opportunities to indirectly infer colony activity and foraging behavior (still need to be added by detecting foraging sounds).

Here and in Introduction, we conservatively replaced "an effective proxy" to "indirectly infer colony activity".

2. Line 166: Add "and satellite images".

We added "aerial images" and referred to Egevang et al. 2003.

3. Line 178: It would be beneficial to provide more specific suggestions for how passive acoustic monitoring of little auk can guide conservation actions, such as identifying key foraging areas or assessing population trends. Additionally, the authors could discuss how involving local communities, such as the Inuit, in the monitoring process could promote sustainable development and support conservation efforts.

We mentioned that passive acoustic monitoring might help in assessing population trends and that inclusive, co-designed work with locals is very valuable.

Methods:

1. Line 185: Remove "to be published elsewhere"

Removed.

2. Line 188: Clarify the statement about all little auks leaving on September 1st to avoid any confusion or overgeneralization.

We have rewritten as "by the beginning of September" to avoid day-scale precision.

3. Line 206: Provide the sensitivity information of the recording system.

Provided.

4. More details on the soundscape analysis could be provided in the manuscript, such as the procedures to generate long-term spectrograms and how the LTSs were used in identifying little auk calls.

In the Methods, we have mentioned that the procedures to generate LTSs were adopted from Guan et al. (2015) and have provided further details on resolution and role of LTSs in sound identification.

Table 1: Includes the authors' names in a separate column, which makes it easier to quickly identify the researchers associated with each study. Additionally, add a short description of the primary finding and survey methods will make the table more informative.

Thank for this advice. Table was extended. Authors' names have been included. As primary findings appear in the caption, to avoid repetition, we added only survey methods.

Figure 2:

1. Change the y-axis label to "Frequency (kHz)" to avoid showing $\times 10^4$. Also, adjust the x-axis limit to exclude periods without observation.

Changed to kHz; periods have been adjusted to avoid mismatch between day ticks.

2. Convert the power spectral density (PSD) plot to a probability distribution to enhance readability.

Converted and marked percentiles.

3. Consider calibrating the recording sensitivity, such as the value listed on the recorder specification, to display the actual dB scale instead of relative values.

The manufacturer, Wildlife Acoustics, does not provide such calibration value for this recorder. On April 27, we contacted the manufacturer, and reconfirmed that such value is not available.

4. Focus on frequencies below 10 kHz and expand the low-frequency range. Use arrows or boxes to highlight the relevant features of the spectrogram.

Y-axis was limited to 10 kHz in long-term spectrograms. PSD and short plots show full spectra to avoid hiding spectral properties of little auk sounds, which can exceed 10 kHz. Relevant frequencies were labelled with lines.

Figure 3:

1. Where is the data recorded from the village site? Exclude periods without observation.

The village data has no clear diel signal, as shown in Fig. 2a and explained in the text. Shortening x-axis is indeed simple. However, we use integer, full-days for ticks as a reference for an eye, because we deal with diurnal signal.

2. You can't put SPL as the y-axis label if you didn't calibrate the recording sensitivity.

This criticism would be valid if we refer to [uPa]. The unit of our y-axis label is [dB], which is a pseudo-unit.

Figure 4. 1. The 1974 data may not be too old for supplement the current observation result. The climate condition and behavior of little auk may be very different.

We suppose the Reviewer meant "may be too old." It is hard to justify exclusion of this data for 2 reasons: (i) it is the only published survey on the diel activity of little auks in Greenland; (ii) hypothetical shifts in behaviour are yet to be shown. Moreover, the data were included as a general evidence of rhythm's existence.

2. Remove redundant panel, consider add more statistical analysis to support your statement.

The redundant panel has been removed. Statistical analysis supporting noise-trend was included into the text and Fig. 3.

Reviewer #3 (Remarks to the Author):

The study seems to examine a diel rhythm of vocal activity of a polar seabird. Diel rhythmicity of various activities both of animals and humans in conditions of continuous daylight remains an intriguing issue. Still not many species has been properly examined in this context, thus mechanisms and taxonomical patterns are not fully recognized.

The study species is truly polar animal (endemic for the Arctic), and so ideal for examining the topic of daily activity patterns in the conditions of polar day. As a matter of fact such an issue has been already considered (specifically colony attendance patterns and foraging activity has been examined in respect to the time of the day). However, the issue has been examined locally and so it is still not clear how universal are the findings then there is still a vast area of various research questions that could be asked.

We note that a diel rhythm of vocal activity of a polar seabird has not been considered so far.

Unfortunately the present study has a serious flaw, and I believe that it cannot be fixed based on the presently used data. First the aim of the study is very much unclear, then whatever the aim is the design of the study is not really adequate. Below I elaborate on this and provide a list of more specific comments on the manuscript content. The latter are also sometimes quite serious (being linked to the major concerns).

We respectfully disagree with such assessment and show below that none of the listed concerns can be called a “serious flaw”. Furthermore, the aims have been clarified.

Major concerns:

1. Unclear aim of the study and so the narration: I am very much confused is the study about diel rhythmicity of the little auk in given site and period? or is it rather about an approach to examine the issue, i.e. with passive acoustic recording. In any case it is also unclear is the study about just birds vocal activity or their colony attendance? Then, at some point a vocal repertoire is mentioned as a study aim. The whole manuscript sounds like a mixture of everything but then applied methodology is not really adequate to any of the issues (see more below). Whatever the aim would be, also its background and interpretation should be elaborated (considering broader ecological/evolutionary context, relevant findings on other species, etc).

We hope major revisions to the text and title have clarified the aim and narration attempting to (1) detect a diel acoustic cycle in the most abundant arctic seabird, and then (2) compare it with existing ethological knowledge for identifying possible biological reasons behind the cycle. There was no intention to show repertoire as the main study aim and we made sure the language is not misleading in this regard. Adequacy of applied methodology is justified below. As advised (and explained in specific replies), background and interpretation were provided with more details; relevant findings on other species were also included.

2. Design of the study: Since the aim of the study is unclear it is also hard to evaluate correctness of the applied methodology. However, whatever the aim would be the study seems to be not properly measuring/sampling the reality.

Soundscape analysis is an emerging field, most developed underwater, where direct observations and absolute animal counts are usually not feasible [e.g., Duarte et al., 2021]. Against this background, dismissal of the study as “not properly measuring/sampling the reality” is surprising. We hope revisions helped to clarify the aims.

For studying diel rhythmicity (no matter whether that would be vocal activity or colony attendance, or whatever) a longer period of time should be considered. Three days of acoustic recording at the end of the chick rearing period represents just an anecdotal data.

We respectfully disagree with this judgment as it contradicts the most relevant literature on avian and biological rhythms. First, rhythmic processes can be identified as such if they are

experimentally observed to persist for at least 1 or 2 cycles, preferably more [Cassone, 2014; Zielinski et al. 2014]. Second, the longest continuous observations of the most recent study on the diel activity of little auks [Wojczulanis-Jakubas et al., 2020], were 48 hours. In our case, the pattern persisted for at least 3 cycles, agreed with all published literature we could find, and thus can not be dismissed as anecdotal.

Selected three days could be very specific, for example due to weather conditions. Then the end of the season (around fledging) has its specificity, especially in the little auk (it is well known that little auk young depart the colony during night hours).

We mentioned that weather conditions could be a problem (e.g., wind), but were not. The area has generally quiet summer conditions. The possibility of night departure of young at the end of the season has been mentioned. Yet, we could not find any quantitative support for this “well known” fact and would appreciate an appropriate literature advice.

Whatever happens during that time is characteristic to that time and cannot be extrapolated to other periods, neither compared with other data, if those have been collected in different breeding stages (as it is the case on Fig 4).

It is not the case on Fig 4, because different breeding stages (e.g., egg-laying, incubation), were not included. Moreover, even in the early incubation stage, previous 1969 study noticed distinct diurnal rhythm, with great activity after midnight.

For evaluating applied methodology, not only control in the non-colony spot (here a village) should be considered but also the control at the colony area (for example, video recording of the birds presence). Again, no matter what is actually considered here - just vocal activity or colony attendance, one has to be controlled for the other. This is because the birds can be in the colony but they may remain quiet (and this is what really happens in the field, although that is only the case when birds are in the colony in low numbers, but this is the part of reality).

We agree that more control is better (e.g., cameras, radars). However, on the one hand, the little auk colony attendance has been repeatedly reported for chick-rearing stage (and described by Reviewer #1 as not something new). On the other hand, it has been repeatedly highlighted that image-based monitoring is difficult for burrow-nesting and other cryptic birds and that there is no single, one-fits-all method to document the “reality”. As we do not attempt bird-density estimation from sound, we believe the Reviewer’s personal observation of quiet birds in low numbers fits into our narrative. Especially, as we carefully worded our main finding (i.e., the post-midnight excitement) in the Abstract.

Besides, more recording spots should be considered as setting the devices in a single spot may give information on this random spot only. If one wants to measure the phenomenon at the colony level, different colony areas should be examined/sampled, to account for potentially biasing local effect.

We agree that more spots is preferable when one can afford more devices. However, previous studies repeatedly shown similar activity rhythmicity at different parts of little auk colonies (Evans, 1981; Wojczulanis-Jakubas et al. 2020).

For vocal repertoire, again one has to keep in mind the breeding stage, as the repertoire may change during the breeding season, and especially fledging time may have its specificity. Then, just pointing out few calls (out of many more in the little auk) without any context (e.g. is the signal emitted in a social interaction, at given breeding circumstances, who is the producer (male/female/chick)) is not really informative.

In the original draft, we mentioned that detailed classification of sounds for finding phenology differences is a promising direction of further research. Also, we stated in Methods that the

rationale for pointing out a few calls was to illustrate that the soundscape was composed of little auk sounds (i.e., not something else, like wind).

Then, other sounds in the colony (like a colony chorus, wing beats) are not really a part of vocal repertoire.

Such sounds were mentioned in the initial main text as a part of a soundscape and not as a part of a repertoire.

Other concerns (of various gravity, presented in “chronological” order):

Lines 23-36: Is this really relevant in the study?

Yes, as it shows how little is known about the soundscape of a little-auk colony.

Lines 37: Not clear what gaps in knowledge this is to fill up (see major comment)

We have clarified that while a lack of terrestrial soundscape research in the tropics has been recognized as a crucial gap, Arctic soundscapes remain nonexistent as a topic.

Line 41-42 “(2) demonstrate that the ambient sound is an efficient proxy to colony activity” – with no proper control for the birds colony attendance and a single recording spot this cannot be demonstrated.

We respectfully disagree. First, documentation of an acoustic presence (of whales, earthquakes, canopy birds, etc) without any direct observations is a giant field of research. Second, in our case this task is less difficult as the attendance cycle has been previous documented (while we do not estimate absolute bird density). Indeed, more spots is preferable, but previous studies shown similar activity rhythmicity at different parts of little auk colonies (Evans, 1981; Wojczulanis-Jakubas et al. 2020).

Line 43: document vocalization repertoire of the species – you present very limited number of calls and without any context it is not really informative (see major comment)

We replied to the major comment above and corrected that we generated the first dataset that can be useful for repertoire analysis. Also, more quantitate analysis has been included.

Line 45: The 56Gb (>144 h) - may sounds impressive but in the context of the study question (whatever it would be) it may not be enough. Besides such a general information is misleading here, as this is not 6 days of continuous recording in given spot, but 3 days in two spots (and that makes a big difference).

Removed.

Lines 45-52: It is unclear what analysis and why was performed here. I do understand that methods are presented at the bottom of the manuscript (journal style) but if so, the results should be presented in a way that one does not have to go to methods to the end of the text to understand what is before, in the results section.

As we have to sit on two chairs and respect both, the journal style and opinions of the Reviewer, we have reiterated the basics of analysis only briefly in Results.

Lines 48-51: Unclear what it is about.

The preceding phrase introduced spectra. The lines 48-51 explain the spectral peaks and their sources. We clarified that the manufacturer’ description referred to the recorder.

Lines 54 “arbitrarily chosen audio files” - why arbitrary? This is not the best approach to examine vocal repertoire of a species.

Our words were not about vocal repertoire, but about the soundscape. They follow an earlier introduced visual information on temporal variation of sound (Fig. 2). To make sure that sounds are indeed coming from birds (when we deal with continuous data exceeding days) arbitrary files work fine, especially since the major pattern persists in almost any file we check (quiet afternoon, loud night).

Lines 56: what is continuous chorus of the colony? – overlapping (various) signals of many individuals? But this cannot really be considered a part of the little auk vocal repertoire

Yes, and a definition has been added. However, our text here does not consider chorus as a part of the vocal repertoire, and instead describes the soundscape at the colony.

Lines 56: wing flapping – is not a part of the little auk vocal repertoire, it may be a part of the acoustic landscape of the little auk colony but without any context is hard to establish its (biological) meaning.

This particular phrase does not label wing flapping as vocal repertoire and has no intention to establish biological meaning. This paragraph describes “what sounds dominated the soundscape at the colony” (which is a synonym to the Reviewer’s “acoustic landscape”).

Lines 51-52 “(suggesting that the vocalizations were insufficient to saturate the ambient noise because the distance involved)” - unclear

We have replaced “saturate” with “dominate” for clarity (correct lines were 61-62).

Figure 3: median-filtered time-series calculated – unclear what does it mean?

Median-filter is a common type of filter. It means that the time-series were smoothed. We have rewritten this for clarity.

Lines 67-68: It is really hard to see this pattern. I mean, one can see the sinusoid but cannot really point out the exact hours.

For addressing this and similar remarks, we added a grid, hourly ticks, and marked the peaks and minima of the noise in Fig. 3.

Lines 69-70: unclear, hard to see the exact duration of the lags, also unclear what “(mean and standard deviation values for six peak/trough pairs)” means

To clarify / help the eye, we added a grid, hourly ticks, and marked six peak/trough pairs (Fig. 3).

Line 71: and what is the reason, why it is important – with three days recording it is just a change not really the pattern.

This is the Results section. A possible reason was provided in the Discussion. Neither in Line 71, nor elsewhere in the paper we call this change as a pattern. Non-stationary analysis with Mann-Kendall test rejects the null hypothesis of trend absence at the alpha significance level of 0.01 (p-value < 0.001).

Figure 4. The breeding period considered here and in the other publications is very different and cannot be really compared that easily. Besides it is unclear how these literature values were extracted (e.g. how number of feedings in given hour were calculated? actually the number of feedings (per nest/colony??) sounds complicated to use here).

Very different breeding periods (e.g., incubation) were not included into our comparison. Still, in the initial text, we carefully acknowledged that comparison is not easy. The values were extracted from published plots; general features of feeds calculations were presented in Supp. Note 1 (we clarified that “per chick”).

Lines 89-92: Unclear

We have broken this long sentence into several and hope it is now easier to read.

Table 1: Counting gulls (any predator) in the colony is tricky as they they adjust their presence to the presence of the prey

This comment refers to the previously published paper by Stempniewicz, 1986; indeed, we agree that counting multiple flying objects with an eye is hard.

Lines 101 “slightly” – rather considerably!

“Considerably” different breeding stages were not included. For instance, from the period before egg-laying to the period of incubation the Little-Auk attendance cycles became shorter and persisted until the end of the season (from about 48 to 24 h; Stempniewicz, 1986).

Lines 101-106: Unclear how the lag was calculated, and presented (one is supposed to guess it/calculate on their own from the fig 3?)

There is no need “to guess it/calculate on their own from the fig 3” in Discussion, because its calculation and value were presented in Results.

Lines 111-113: Hard to see the logic behind. You cannot accept one of the hypotheses (that requires a separate testing) just excluding the others (which you actually have not tested properly).

Discussion section introduces conflicting hypotheses in the literature and reveals the lack of coherent interpretation for the diel activity cycle of a little auk. Therefore, it is logical to pay attention to details which help to assess adequacy of published biological interpretations. The logic of the paragraph is: a well-known commuting may cause a lag, however a lag in a hypothetical predator’s pressure has no support in literature.

Lines 191-192: It does not mean anything. You could not hear it simply because of being to far/at different time/etc. Not hearing the birds vocalizing at sea (while not really measuring it properly) you can not conclude they do not vocalize while at sea.

As answered to a similar remark by Reviewer #1, we work with local Inuit hunters in the sea since 2015. Hunting and our observations on marine mammals in open water required silence, switched-off engine, low wind [Podolskiy and Sugiyama, JGR, 2020], and often corresponded to submergence into flocks of little auks, at night and day. Indeed, we work towards eliminating subjective experience from animal observations, yet the contrast between the colony and the sea is so stunning that we prefer to keep it. We explained this and replaced “heard” with “noticed”.

Lines 201-203: What format were you recording?

This was initially shown in the next sentence (i.e., *.wav).

Lines 210-211 – unclear why you constructed the two types of the spectrograms and what is the difference between them.

We did not construct two types of the spectrograms. Perhaps, the Reviewer refers to LTS and SPL, so we clarified that the former is a 3D-type of data and the latter is 2D data, which is easier to compare with other time-series.

Lines 213-214: Unclear

We replaced “granular consideration of discrete acoustic events” with “detailed consideration of different sound types” and hope it is better now.

Once again, we thank the Reviewers for their help to improve our manuscript.

Reviewers' comments:

Reviewer #2 (Remarks to the Author):

I recognize the efforts made by the authors in revising the manuscript and addressing questions and comments raised by reviewers. However, the revised manuscript still presents some areas of concern. The authors highlight the central scientific question as understanding the diel activity of the little auk in a consistently lit summer Arctic environment. Unfortunately, as the other two reviewers have noted, the data presented is based on field recordings from only two locations and spanning just a few days. Only one of these locations is close to a colony of the little auk. This limited scope makes it challenging for me to grasp the acoustic activity of the little auk and what biotic or abiotic drivers behind the observed acoustic activity pattern.

Firstly, the authors utilize the technique of soundscape sensing, and use the median sound intensity as an indicator of little auk acoustic activity. The soundscape, however, consists of both biotic and abiotic sounds from the study area. Given our limited understanding of the Arctic soundscape, it is hard to predict how various sound sources impact soundscape dynamics. The authors attempt to use Figure 2 to explain the relationship between ambient sound spectra and little auk sounds, but the LTS lacks signals that resemble the provided example spectrograms. Furthermore, based on the updated repertoire analysis in the final discussion section, only trilling calls have a mean frequency of 2 kHz, but other signals show diverse mean frequencies. Without detailed analysis explaining what contributes to the 2 kHz peak, the use of sound intensity as a measure of little auk acoustic activity remains questionable.

In the revised manuscript, the authors have applied detection and classification algorithms to analyze the acoustic activity of the little auk. Yet, their findings are briefly discussed in the final discussion section and omitted from the results section entirely. This, coupled with the manuscript's complex structure, as noted by other reviewers, makes it exceedingly difficult to comprehend the authors' main points.

In addition, the limited recording duration and single colony focus is insufficient for investigating the circadian-like rhythm in little auk sound activity (not sure if you can say this is a hypothesis). Expanding the recording duration to examine changes in acoustic activity under different light conditions and discussing potential drivers could be one solution. Another could be to increase the number of studied sites and colonies. This is certainly possible given the compact size of commercially available recorders.

Lastly, I would urge the authors to refrain from using sound pressure levels (SPL) without a calibrated microphone. Indeed, an SPL can register a negative value if the recorded sound pressure is exceedingly low, and I doubt that the recorder used in this study or the quietness of the study area can achieve such sensitivity. Without a sensitivity calibration, the authors are measuring relative sound intensity, not SPL. Although these intensities can be represented on a dB scale, they should not be referred to as SPL. This is but one example in the rebuttal letter where the authors appear to resist the suggestions from reviewers. I strongly advise the authors to thoughtfully consider these recommendations. I believe doing so will reveal the collective aim of all three reviewers, which is to elevate the quality of this intriguing research.

Reviewer #4 (Remarks to the Author):

I've read the revised manuscript with a main emphasis on questions raised by earlier reviewers. I also worked through the response letter, feeling somewhat uneasy because of the palpable tension: I believe that we should keep calm and remain on solid scientific ground. My overall impression is that the manuscript offers very interesting new data and in-depth analyses. I think that the revision

satisfies many concerns raised by the reviewers, and, in my view, even where problems could not be resolved, the paper fits with my perception of the field. Below I list some small issues that I think need to be addressed.

Small issues:

Abstract: "likely parallels" still overstates the evidence. Why not simply say "we interpret this pattern as reflecting foraging activities"

I would also modify the first sentence. Even for the dawn chorus, answers are not fully clear. There are several competing hypotheses, recently nicely reviewed by Gil and Llusia (2020). It's certainly not easy to show the main driver of a rhythm, because, as mentioned by the authors, rhythmic behaviour is fundamental for many processes. I think this paper is stronger if it focused on its own system (polar) from the start.

p. 2 after "However": "it has rarely been investigated in polar ..." is wrong unless authors specify "by acoustic methods". The authors themselves now cite several further avian studies, e.g. on Murres and penguins, and there are several others.

Just afterwards, on the dawn chorus, see the comment to the abstract.

p. 2-3: the authors state that the described calls may be one and the same. I couldn't find what they concluded on that question in the Discussion?

p. 3 Results: can you please state the time of recording (July or August) when mentioning the two sites? It later gets a bit confusing when you talk about July and August data. Can you please also explain how this time step between recordings relates to the chick developmental stages? This is important for interpreting July-August differences.

p. 3: "several arbitrarily chosen" -> how many?

p. 6: leave out the bracketed sentence "This fact ..." - as mentioned above, there is uncertainty even for the dawn chorus. Thereafter, "two ... hypotheses". There are more - you later discuss fledging, as a third one, and there may still be more. So please rephrase more cautiously and also here already mention fledging.

p. 17: "Wojczulanis-Jakubas ... also detected ..." "diurnal" is wrong - it would mean day-active, but this can't be the case if rhythms deviate from 24 h. "Diel" is also wrong since it refers to a 24 h cycle, so this change did not solve the problem. It is interesting that the little auks deviated from 24 h. I would talk about "a circadian rhythm" because it might well free-run under continuous polar light (see also work by Bulla et al., e.g., 2016). This possibility should be mentioned also in the main paper. Free-running is indeed the third of 3-4 main rhythmic types, next to diel and non-rhythmic, and possibly tidal. On this point, I agree with the previous reviewer 1, although your words of caution for the little auk study are appreciated and could be added.

Signed Barbara Helm

Submitted: 16 May 2023

Reviews: 17 Jul 2023

Dear Editors and Reviewers,

Thank you for evaluating our revised manuscript and suggesting further ways of strengthening it. Our detailed point-by-point responses are provided below (shown in **bold**; modifications are highlighted in the manuscript; to ease responding, we took liberty to split some long paragraphs). We are pleased to see how the reviews have elevated the quality and depth of the paper. With apologies for this unfortunate delay (all authors were out of office for a few months).

Yours sincerely,
Evgeny Podolskiy
and co-authors

Reviewers' comments:

Reviewer #2 (Remarks to the Author):

I recognize the efforts made by the authors in revising the manuscript and addressing questions and comments raised by reviewers. However, the revised manuscript still presents some areas of concern. The authors highlight the central scientific question as understanding the diel activity of the little auk in a consistently lit summer Arctic environment. Unfortunately, as the other two reviewers have noted, the data presented is based on field recordings from only two locations and spanning just a few days. Only one of these locations is close to a colony of the little auk. This limited scope makes it challenging for me to grasp the acoustic activity of the little auk and what biotic or abiotic drivers behind the observed acoustic activity pattern.

We have added 3 months of quantitative data on attendance patterns, which further support all our claims (Fig. 4b and Supplementary Fig. 4, as shown below). In regard to the limited scope: this question might be prone to subjective judgment in the absence of previous research. This, however, is not the case, because we are guided by previous literature on circadian rhythms (as detailed also below). First, to confirm a rhythm one needs to observe at least one-two cycles, as we did. Second, simultaneous studies on a few little auk subcolonies, suggested that the diel activity was similar. Against this background, and considering that our objective is to show the existence of behaviour-related acoustic cycle, the data is adequate to the purpose. Also, we show that the observed acoustic activity pattern is clearly biotic.

Firstly, the authors utilize the technique of soundscape sensing, and use the median sound intensity as an indicator of little auk acoustic activity. The soundscape, however, consists of both biotic and abiotic sounds from the study area. Given our limited understanding of the Arctic soundscape, it is hard to predict how various sound sources impact soundscape dynamics.

This foremost remark is of little concern in our study. Because we used several principally different approaches to re-confirm that soundscape is certainly biotic (i.e., listening, vocalization detections, biophony indices, and alignment with known bio-rhythms).

The authors attempt to use Figure 2 to explain the relationship between ambient sound spectra and little auk sounds, but the LTS lacks signals that resemble the provided example spectrograms.

Please note that the time resolution of the LTS (10 s) is too coarse for direct comparisons with the short examples (e.g., Figs. 2e&2f are 2 s long, which is below the temporal resolution of the LTS). Nevertheless, the elevated energy of short calls still can be seen as broad-band lines in the spectrogram (Fig. 2c).

Furthermore, based on the updated repertoire analysis in the final discussion section, only trilling calls have a mean frequency of 2 kHz, but other signals show diverse mean frequencies. Without detailed analysis explaining what contributes to the 2 kHz peak, the use of sound intensity as a measure of little auk acoustic activity remains questionable.

First, the trilling call is the most common and the longest call of the species (Ferdinand, 1969; Osiecka et al., 2023). Second, the 2 kHz peak was confirmed aurally as bird chorus. Third, other signals were broad-band, and even if had higher mean frequency, will loose it with distance, leaving the longest, low-frequency sources more impactful in a long-term. For these reasons, in combination with alternative verification (biophony indices and detections), the use of intensity is adequate. We elaborated further in this regard (lines 255 - 257).

In the revised manuscript, the authors have applied detection and classification algorithms to analyze the acoustic activity of the little auk. Yet, their findings are briefly discussed in the final discussion section and omitted from the results section entirely. This, coupled with the manuscript's complex structure, as noted by other reviewers, makes it exceedingly difficult to comprehend the authors' main points.

The referred findings were not “omitted from the results section entirely,” but were mentioned at the end of the section (p. 4-5). To improve clarity, we have added further details on the outline of the section and detections (lines 69-76, 117-122). We also humbly remind that: (1) additional analysis of repertoire was not our idea, but the reviewers' request (it remains supplementary, as it does not adjust the main points, but confirms the biotic nature of the soundscape). (2) Any notes by other reviewers have been answered and addressed in our previous rebuttal letter.

In addition, the limited recording duration and single colony focus is insufficient for investigating the circadian-like rhythm in little auk sound activity (not sure if you can say this is a hypothesis). Expanding the recording duration to examine changes in acoustic activity under different light conditions and discussing potential drivers could be one solution. Another could be to increase the number of studied sites and colonies. This is certainly possible given the compact size of commercially available recorders.

We have added 3 years of quantitative attendance data from another colony nearby, which indirectly, but strongly, support all our interpretations (Fig. 4b and Supplementary Fig. 4, lines 351-359). However, we respectfully disagree that acoustic data is insufficient. First, formal documentation of a circadian rhythm requires at least one or two cycles (Cassone, 2014), and - comparing to previous studies based on shorter 48 h data (Wojczulanis-Jakubas et al., 2020) - our data is adequate. Second, to date, to our knowledge, there has been no little-auk-diel-activity paper focused on multiple colonies due to limited resources. Moreover, it was shown that in general, subcolonies show synchrony, especially towards fledging (Evans, 1981; Stempniewicz, 1986). Indeed, we agree that bringing long-term recorders to multiple sites is an exciting proposal. However, realistically speaking, for reaching some of the most remote areas of Greenland (7 flights from Sapporo to Siorapaluk) one needs substantial funding, human resources, and time for such major undertaking, which we hope to pursue in the future.

Lastly, I would urge the authors to refrain from using sound pressure levels (SPL) without a calibrated microphone. Indeed, an SPL can register a negative value if the recorded sound pressure is exceedingly low, and I doubt that the recorder used in this study or the quietness of the study area can achieve such sensitivity. Without a sensitivity calibration, the authors are measuring relative sound intensity, not SPL. Although these intensities can be represented on a dB scale, they should not be referred to as SPL. This is but one example in the rebuttal letter where the authors appear to resist the suggestions from reviewers. I strongly advise the authors to thoughtfully consider these recommendations. I believe doing so will reveal the collective aim of all three reviewers, which is to elevate the quality of this intriguing research.

Previously we have explained that our sound levels are relative (because dB is a pseudo-unit). To avoid misunderstanding, we have corrected SPL as a relative sound intensity, RSI (Fig. 3, lines 101, 363-364). All other remarks by the four reviewers have been carefully answered or addressed in this and previous rebuttal letters.

Reviewer #4 (Remarks to the Author):

I've read the revised manuscript with a main emphasis on questions raised by earlier reviewers. I also worked through the response letter, feeling somewhat uneasy because of the palpable tension: I believe that we should keep calm and remain on solid scientific ground. My overall impression is that the manuscript offers very interesting new data and in-depth analyses. I think that the revision satisfies many concerns raised by the reviewers, and, in my view, even where problems could not be resolved, the paper fits with my perception of the field. Below I list some small issues that I think need to be addressed.

Thank you very much for this positive appraisal, and your suggestions, which we have followed.

Small issues:

Abstract: "likely parallels" still overstates the evidence. Why not simply say "we interpret this pattern as reflecting foraging activities"

We're happy to clarify as suggested.

I would also modify the first sentence. Even for the dawn chorus, answers are not fully clear. There are several competing hypotheses, recently nicely reviewed by Gil and Llusia (2020). It's certainly not easy to show the main driver of a rhythm, because, as mentioned by the authors, rhythmic behaviour is fundamental for many processes. I think this paper is stronger if it focused on its own system (polar) from the start.

Thank you for pointing this out. We're happy to make it clear that even the dawn chorus is hard to explain, and to include this new reference into our bibliography list.

p. 2 after "However": "it has rarely been investigated in polar ..." is wrong unless authors specify "by acoustic methods". The authors themselves now cite several further avian studies, e.g. on Murres and penguins, and there are several others.

Specified as suggested (line 20).

Just afterwards, on the dawn chorus, see the comment to the abstract.

We adjusted this phrase accordingly and added the suggested reference (line 22).

p. 2-3: the authors state that the described calls may be one and the same. I couldn't find what they concluded on that question in the Discussion?

We rewrote this (lines 45-47); matching calls to phonetics of different languages wasn't our aim.

p. 3 Results: can you please state the time of recording (July or August) when mentioning the two sites? It later gets a bit confusing when you talk about July and August data. Can you please also explain how this time step between recordings relates to the chick developmental stages? This is important for interpreting July-August differences.

As suggested, we have clarified the time of recording and the phenological stage (lines 66-67). Yet, we recall that the Jul-Aug difference was mainly due to the distance.

p. 3: “several arbitrarily chosen” -> how many?

We indicated that around 15 (line 84).

p. 6: leave out the bracketed sentence “This fact ...” - as mentioned above, there is uncertainty even for the dawn chorus. Thereafter, “two ... hypotheses”. There are more - you later discuss fledging, as a third one, and there may still be more. So please rephrase more cautiously and also here already mention fledging.

The bracketed sentence was removed as suggested. Here, we are happy to rephrase that there are “at least three ... hypotheses” and to mention fledging hypothesis (lines 148-149).

p. 17: “Wojczulanis-Jakubas ... also detected ... “diurnal” is wrong - it would mean day-active, but this can't be the case if rhythms deviate from 24 h. “Diel” is also wrong since it refers to a 24 h cycle, so this change did not solve the problem. It is interesting that the little auks deviated from 24 h. I would talk about “a circadian rhythm” because it might well free-run under continuous polar light (see also work by Bulla et al., e.g., 2016). This possibility should be mentioned also in the main paper. Free-running is indeed the third of 3-4 main rhythmic types, next to diel and non-rhythmic, and possibly tidal. On this point, I agree with the previous reviewer 1, although your words of caution for the little auk study are appreciated and could be added.

Thank you for carefully considering this point. While “diel” was used by Wojczulanis-Jakubas et al. (2020) in their title, the usage of “circadian” might be criticised as it is defined as a process which persists for at least 1 or 2 cycles when the animal is placed in ideal constant light conditions (Cassone, 2014). As a compromise, we rephrased as a “semi-diel”, which is commonly used in ethology (end of p.18). The possibility of the free-running and the suggested reference are now cited in the paper (lines 23-24, p. 19, Bulla et al. 2016).

Once again, we thank the Reviewers for their help to improve our manuscript.

REVIEWERS' COMMENTS:

Reviewer #4 (Remarks to the Author):

From reading the materials, I can confirm that the authors have made all the effort (and more) than can be expected for a revision. I think this manuscript now makes a fine and interesting contribution to the broader fields on which it touches.

REVIEWERS' COMMENTS:

Reviewer #4 (Remarks to the Author):

From reading the materials, I can confirm that the authors have made all the effort (and more) than can be expected for a revision. I think this manuscript now makes a fine and interesting contribution to the broader fields on which it touches.

Many thanks for your final review and all the help with our manuscript!